# Layer-specific optogenetic activation of pyramidal neurons causes beta–gamma entrainment of neonatal networks

Sebastian H. Bitzenhofer[1,*], Joachim Ahlbeck[1,*], Amy Wolff[1], J. Simon Wiegert[2], Christine E. Gee[2], Thomas G. Oertner[2] & Ileana L. Hanganu-Opatz[1]

Coordinated activity patterns in the developing brain may contribute to the wiring of neuronal circuits underlying future behavioural requirements. However, causal evidence for this hypothesis has been difficult to obtain owing to the absence of tools for selective manipulation of oscillations during early development. We established a protocol that combines optogenetics with electrophysiological recordings from neonatal mice *in vivo* to elucidate the substrate of early network oscillations in the prefrontal cortex. We show that light-induced activation of layer II/III pyramidal neurons that are transfected by *in utero* electroporation with a high-efficiency channelrhodopsin drives frequency-specific spiking and boosts network oscillations within beta–gamma frequency range. By contrast, activation of layer V/VI pyramidal neurons causes nonspecific network activation. Thus, entrainment of neonatal prefrontal networks in fast rhythms relies on the activation of layer II/III pyramidal neurons. This approach used here may be useful for further interrogation of developing circuits, and their behavioural readout.

[1] Developmental Neurophysiology, Institute of Neuroanatomy, University Medical Center Hamburg-Eppendorf, 20251 Hamburg, Germany. [2] Institute for Synaptic Physiology, University Medical Center Hamburg-Eppendorf, 20251 Hamburg, Germany. * These authors contributed equally to this work. Correspondence and requests for materials should be addressed to I.L.H.-O. (email: hangop@zmnh.uni-hamburg.de) or to S.H.B. (email: sebbitz@zmnh.uni-hamburg.de).

The developing brain functions according to unique processing rules. The highly discontinuous and fragmented temporal organization of the network activity[1–4], the depolarizing action of GABA (γ-aminobutyric acid)[5,6] as well as the patterns of local and long-range network coupling[7,8] relying on specific neuronal subpopulations and their transient connectivity[9,10] profoundly differentiate the developing brain from the adult one. The mechanisms underlying the unique wiring and function of the brain early in life are still poorly understood. It is also still a matter of debate whether the development-specific activity patterns are critical for the adult function and behavioural performance or simply a by-product of maturation processes.

In adulthood, brain function is tightly related to the activity of neuronal circuits. Pyramidal neurons and several classes of inhibitory interneurons dynamically interact to generate network activity in distinct frequency bands and enable diverse behaviours[11,12]. Resolving these circuits by identifying the contribution of each neuronal population to the rhythmic network activity and overall brain function in vivo has been recently enabled by the development of technologies to specifically control and manipulate neuronal activity at fast timescales[13,14]. Activation or suppression of action potentials (APs) in distinct neuronal populations by artificial incorporation of diverse light-sensitive proteins has been utilized to decipher the underlying mechanisms of memory[15–18], sensory and multisensory processing[19–21], and neuropsychiatric disorders[22–24].

Similar interrogation of developing circuits is currently missing. Consequently, the origin of neonatal brain rhythms remains unknown. Correlative evidence emphasized the contribution of complex and precisely tuned cellular interactions to the emergence of discontinuous patterns of early oscillatory activity[2,3,8,25,26], but induction of a distinct network state by cell-type-specific activation was not possible.

Here, we introduce a protocol for causal manipulation of neuronal and network activity in the neonatal brain (that is, postnatal day (P) 8–10) in vivo using targeted optogenetic stimulation. For this, cell-type- and layer-specific transfection of neurons by in utero electroporation (IUE) was combined with head-fixed recordings of local field potential (LFP) and spiking activity in neonatal mice during light stimulation. By these means we identified pyramidal neurons in layer II/III but not layer V/VI of the prelimbic subdivision (PL) of the prefrontal cortex (PFC) as key elements for the generation of beta–gamma oscillations during neonatal development.

## Results

**Cell-type-specific transfection during early development.** The first prerequisite for the interrogation of developing circuits with optogenetics is the selective and effective transfection of neuronal populations with light-sensitive proteins early in life. For this, it is necessary to identify a suitable method of transfection, type of promoter giving the best expression and a light-sensitive protein. The most effective strategy for achieving functional expression of optogenetic tools in the adult brain, viral expression systems, cannot be reliably used for the investigation of developmental networks, because it usually requires 1–3 weeks until the gene expression reaches functional levels and layer specificity is rather poor[27,28]. Our pilot investigation confirmed these findings. Viral vectors based on adeno-associated virus 8 or canine adenovirus that have been described as enabling fast (48 h–6 days) expression in vitro[29,30] led to insufficient, if any, expression of channelrhodopsins (ChRs) in the PL of the PFC from P5–10 mice when injected in vivo 1–2 days after birth. Moreover, the promoters suitable for targeting neuronal subpopulations in adult mice often lack specificity or promote weak and unstable expression during early development.

To achieve stable gene expression in specific subpopulations of neurons in the neonatal brain, we used IUE as it enables cell-type-, layer- and area-specific transfection of neurons already prenatally without the need of cell-type-specific promoters of a sufficiently small size[31–33]. Pyramidal neurons either in layer II/III or in layer V/VI were addressed by this approach, since our previous investigations identified their activity as temporal correlate of network oscillations in neonatal PL[8,34]. Aiming at targeting pyramidal neurons in layers II/III of the PL, we injected constructs coding for the highly efficient fast-kinetics double mutant ChR2 E123T T159C (ET/TC)[35] and the red fluorescent protein tDimer2 into the right lateral ventricle at embryonic day (E) 14.5–15.5. Subsequenty, we applied an electrical field to the embryo's head to transfect neural precursor cells in the subventricular zone (Fig. 1a,b). This protocol is intended to selectively transfect pyramidal neurons in upper cortical layers (layer II/III) due to the distinct timing, origin and migration paths (that is, radial vs. tangential) of cortical pyramidal neurons and interneurons[36]. The procedure for targeting pyramidal neurons in layer V/VI was similar in its settings (for example, angle, electrical field), but performed at E12.5. To reach a high transfection rate, we tested the efficiency of several promoters in the neonatal brain. While human elongation factor 1α (EF1α) and synapsin promoters led to few or no tDimer2-positive neurons in the PL, the cytomegalovirus enhancer fused to chicken β-actin (CAG) promoter led to a robust expression in both upper and deeper layers that remained constant throughout the entire period of postnatal development (P5–25).

Analysis of consecutive coronal sections from IUE-transfected P8–10 mice revealed that tDimer2-positive neurons are mainly present in the PL, cingulate and infralimbic cortices and to a lesser extent in motor cortex (2 out of 9 for IUE at E15.5 and 2 out of 12 for IUE at E12.5) (Fig. 1c,f and Supplementary Fig. 8a). Staining for NeuN showed that $34.7 \pm 0.8\%$ ($n = 7$ pups) of neurons in prelimbic layer II/III and $33.1 \pm 1.2\%$ ($n = 5$ pups, $P = 0.288$) of neurons in layer V/VI were transfected by IUE at E15.5 or E12.5, respectively. The pyramidal-like shape and orientation of primary dendrites as well as the expression of $Ca^{2+}$/calmodulin-dependent protein kinase II (CaMKII) and the absence of positive staining for GABA confirmed that the expression constructs are exclusively integrated into cell lineages of pyramidal neurons (Fig. 1d,e,g,h). In line with the timing of migration[37], IUE at E15.5 selectively targeted neurons in the upper prelimbic layers II/III ($99.3 \pm 0.2\%$). Only $<0.5\%$ of tDimer2-positive neurons were detected in layer I or deep layers V/VI. Targeting of neurons in layer V/VI was less precisely confined, due to ongoing migration of a small fraction of neurons transfected at early age. However, the majority of transfected neurons were located in layer V/VI ($87.7 \pm 0.9\%$). Omission of ChR2(ET/TC) from the expression construct (that is, opsin-free) yields similar expression rates and distribution of tDimer2-positive neurons (Fig. 1e,h). Moreover, the success rate of transfection by IUE was similar in both groups of mice (Supplementary Fig. 1a,d).

To exclude nonspecific effects of the transfection procedure by IUE on the overall development of animals, we assessed the developmental milestones and reflexes of electroporated opsin-expressing and opsin-free mice as well as of non-electroporated mice (Supplementary Fig. 1). While IUE caused significant reduction of litter size (non-electroporated: $7.7 \pm 0.3$ pups per litter; IUE: $5 \pm 0.2$ pups per litter; $P = 0.004$), all investigated pups had similar body length, tail length and weight during the early postnatal period. Vibrissa placing, surface righting and cliff aversion reflexes were also not affected by IUE or transfection of

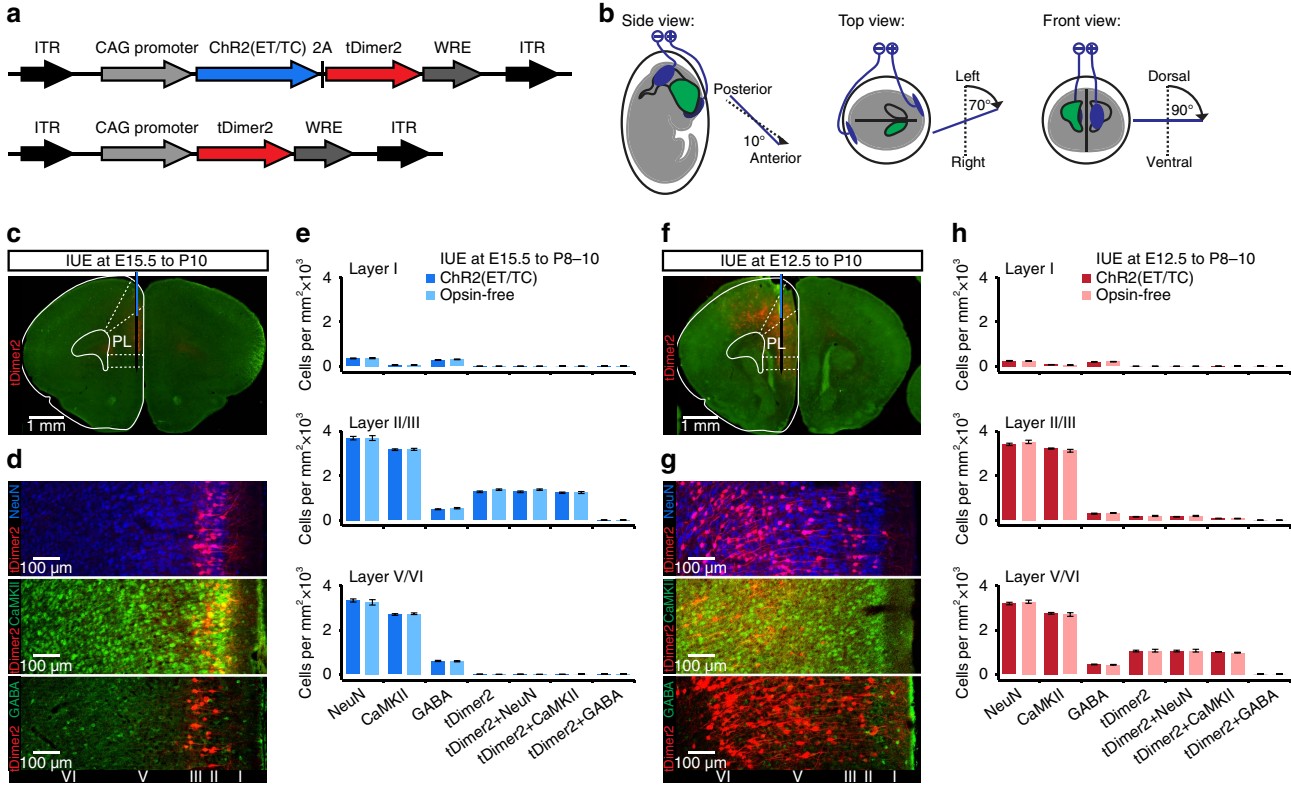

**Figure 1 | Cell- and layer-specific transfection of neonatal prelimbic cortex by site-directed *in utero* electroporation.** (**a**) Structure of the ChR2(ET/TC)-containing and opsin-free constructs. (**b**) Schematic drawing illustrating the orientation of electrode paddles for specific targeting of pyramidal neurons in layer II/III and V/VI of PL. (**c**) tDimer2-expressing cells (red) in a 50-μm-thick coronal section of a P10 mouse at the level of the prefrontal cortex after IUE at E15.5. (**d**) Photographs displaying NeuN (blue), $Ca^{2+}$/calmodulin-dependent protein kinase II (CaMKII) (green) and GABA (green) immuno-histochemistry in relationship to tDimer2 expression (red) at P10 after IUE at E15.5. Note that the transfection is restricted to CaMKII-positive and GABA-negative neurons in layer II/III. (**e**) Bar diagrams displaying the mean density of NeuN-, CaMKII-, GABA-, tDimer2-, tDimer2 + NeuN-, tDimer2 + CaMKII- as well as tDimer2 + GABA-positive cells in layer I (top), II/III (middle) and V/VI (bottom) of PL from P8 to P10 mice after IUE with constructs containing CAG-ChR2(ET/TC)-2A-tDimer2 (dark blue bars, $n = 28$ slices from 7 pups for tDimer2, NeuN, GABA; $n = 20$ slices from 5 pups for CaMKII) or opsin-free constructs (light blue bars, $n = 28$ slices from 7 pups for tDimer2, NeuN, GABA; $n = 20$ slices from 5 pups for CaMKII). (**f–h**) Same as (**c–e**) for IUE at E12.5 with constructs containing CAG-ChR2(ET/TC)-2A-tDimer2 (dark red bars, $n = 20$ slices from 5 pups for tDimer2, NeuN, GABA; $n = 20$ slices from 5 pups for CaMKII) or opsin-free constructs (light red bars, $n = 20$ slices from 5 pups for tDimer2, NeuN, GABA; $n = 16$ slices from 4 pups for CaMKII). Note that the transfection is restricted to CaMKII-positive and GABA-negative cells mainly located in layer V/VI. Data are presented as mean ± s.e.m. ITR, inverted terminal repeat. WRE, woodchuck hepatitis virus posttranscriptional regulatory element.

neurons with opsins. These data indicate that the overall somatic development during embryonic and postnatal stage of ChR2 (ET/TC)-transfected mice is normal.

**Light-induced spiking of neonatal neurons *in vitro*.** We first assessed the efficiency of light stimulation in evoking APs in neonatal neurons. For this, whole-cell patch-clamp recordings were performed from tDimer2-positive pyramidal neurons in layer II/III ($n = 14$ cells) and layer V/VI ($n = 12$ cells) in coronal slices containing the PL from P8 to 10 mice after IUE at E15.5 and E12.5, respectively (Figs 2a and 3a). In line with the previously reported 'inside-out' pattern of cortical maturation[38], layer II/III and V/VI pyramidal neurons significantly differed in their passive membrane properties such as resting membrane potential (RMP) (layer II/III $-73.3 \pm 1.8$ mV; layer V/VI $-65.8 \pm 2.3$ mV; $P = 0.012$) and input resistance (layer II/III $1,067.5 \pm 135.0$ MΩ; layer V/VI $614.3 \pm 98.6$ MΩ; $P = 0.024$). These data confirm the more mature profile of neurons in deep layers of the PL compared with superficial layers. All investigated neurons fired overshooting APs in response to sustained depolarization by intracellular current injection

(Figs 2b and 3b). The passive and active properties of ChR2(ET/TC)-transfected neurons were similar to those previously reported for age-matched rats[8]. However, the active properties of layer II/III pyramidal neurons differed from those of layer V/VI neurons (AP threshold: layer II/III $-38.2 \pm 1$ mV, layer V/VI $-43.2 \pm 0.9$ mV, $P = 0.002$; AP half-width: layer II/III $3.2 \pm 0.2$ ms, layer V/VI $2.3 \pm 0.3$ ms, $P = 0.013$). To gain insights whether early oscillations in the neonatal PL are the result of intrinsic neuronal properties or of network activation, we tested the resonance profile of transfected neurons. The impedance profile in response to chirp current injection with linearly increasing frequency from 1 to 50 Hz did not indicate a preferred membrane resonance frequency of layer II/III and V/VI pyramidal neurons (Figs 2c and 3c).

Blue light pulses (473 nm, 5.2 mW mm$^{-2}$) depolarized all fluorescently labelled neurons, yet the suprathreshold activation critically depended on the promoter used as well as the intensity and duration of light pulses. Consistent with the rather weak fluorescence of neurons transfected with ChR2(ET/TC) under the control of EF1α, AP firing was evoked when the duration of light pulses exceeded 100 ms. In contrast, the use of the CAG promoter resulted in reliable firing of single APs in response to 3-ms-long

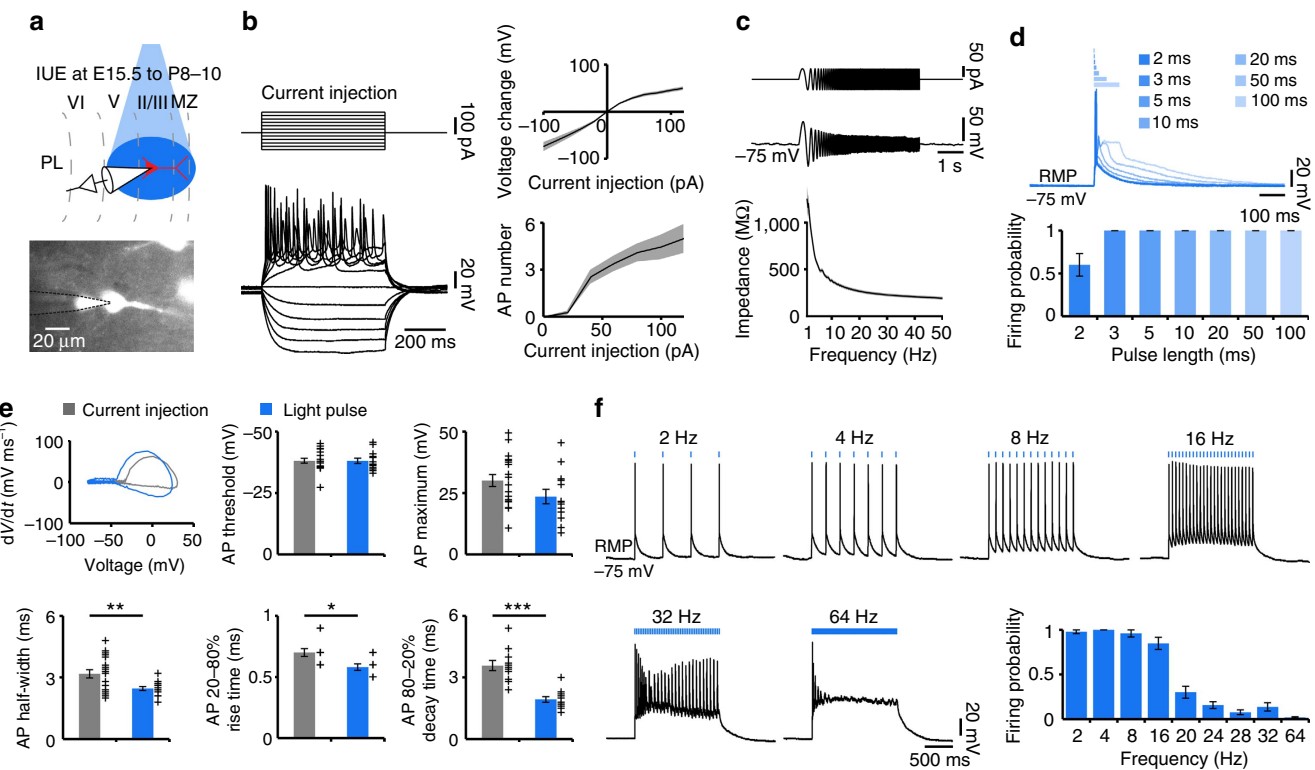

**Figure 2 | Optogenetic activation of prelimbic layer II/III pyramidal neurons *in vitro*.** (**a**) Experimental approach for combined light stimulation and whole-cell recordings from IUE-transfected layer II/III pyramidal neurons in coronal slices of P8–10 mice. (**b**) Left, voltage responses of a representative neuron to hyper- and depolarizing current pulses. The RMP was −73.7 mV. Right, averaged plots of the current–voltage relationship (top) and current-firing rate relationship (bottom) ($n = 20$ neurons). (**c**) Top, waveform of subthreshold chirp current injection with increasing frequency from 1 to 50 Hz. Middle, voltage response of a representative neuron. Bottom, averaged plots of the impedance ($n = 16$ neurons). (**d**) Top, voltage responses of a representative ChR2(ET/TC)-transfected neuron to blue light pulses (473 nm, 5.2 mW mm$^{-2}$) of 1 (dark blue) to 100 ms (light blue) duration. Bottom, bar diagram displaying the mean firing probabilty of transfected neurons in response to blue light pulses of variable duration ($n = 14$ neurons). (**e**) Top, representative phase plot of APs elicited in a transfected neuron by depolarizing current injection (grey) or a 3-ms-long light pulse (473 nm, 5.2 mW mm$^{-2}$, blue). Bottom, bar diagrams displaying the mean active membrane properties after depolarizing current injections (grey, $n = 18$ neurons) and after light stimulation (blue, $n = 14$ neurons). Individual values are displayed as black crosses. (**f**) Representative voltage responses of a transfected neuron to repetitive trains of 3-ms-long light stimuli at variable frequencies. Bar diagram displaying the mean firing probability of transfected neurons in response to repetitive light stimuli ($n = 14$ neurons). Data are presented as mean ± s.e.m. *$P < 0.05$, **$P < 0.01$ and ***$P < 0.001$, two-sided *t*-tests. MZ, marginal zone.

light pulses (Figs 2d and 3d). APs evoked by light and by current injection were similar in their threshold (layer II/III—light triggered: −38.1 ± 1.1 mV, current triggered: −38.2 ± 1 mV, $P = 0.98$; layer V/VI—light triggered: −41.5 ± 1 mV, current triggered: −43.2 ± 0.9 mV, $P = 0.22$) and peak voltage (layer II/III—light triggered: 23.6 ± 2.9 mV, current triggered: 30.1 ± 2.5 mV, $P = 0.09$; layer V/VI—light triggered: 34.3 ± 5 mV, current triggered: 37 ± 5.1 mV, $P = 0.72$) (Figs 2e and 3e), indicating that light stimulation causes physiological activation of ChR2(ET/TC)-transfected neurons. However, half-width of light-triggered APs was significantly reduced for layer II/III (current triggered 3.2 ± 0.2 ms; light triggered 2.5 ± 0.1 ms; $P = 0.009$), but not for layer V/VI (current triggered 2.3 ± 0.3 ms; light triggered 2.0 ± 0.2 ms; $P = 0.358$) pyramidal cells (Figs 2e and 3e). To mechanistically explain these differences, we modelled the effect of Na$^+$/K$^+$ conductances on the AP time course with the Hodgkin–Huxley model[39] (Supplementary Fig. 2). Modelled AP half-width was modulated by the strength of the current injection for neurons with low (that is, immature), but not high (that is, mature) Na$^+$/K$^+$ conductances, suggesting a strong inward current after stimulation. A similar dependence was experimentally detected for

current injections only for layer II/III pyramidal neurons (Supplementary Fig. 2f).

Precise induction of APs over a broad range of frequencies, corresponding to those of neonatal network oscillations, requires the use of ChRs with fast kinetics that deliver large photocurrents. Adult ChR2(ET/TC)-expressing neurons have been reported to reliably follow stimulation frequencies up to 60 Hz (ref. 35). We stimulated neonatal ChR2(ET/TC)-expressing neurons with trains of 3-ms-long light pulses at frequencies ranging from 2 to 64 Hz. Robust firing was evoked in all neurons throughout stimulation up to 32 Hz, but the probability of triggering APs by trains of light stimuli decreased with increasing stimulation frequency (2 Hz: layer II/III 0.97 ± 0.02, layer V/VI 1 ± 0, $P = 0.328$; 4 Hz: layer II/III 1 ± 0, layer V/VI 0.94 ± 0.06, $P = 0.289$; 8 Hz: layer II/III 0.96 ± 0.04, layer V/VI 0.80 ± 0.09, $P = 0.103$; 16 Hz: layer II/III 0.85 ± 0.07, layer V/VI 0.46 ± 0.11, $P = 0.004$; 20 Hz: layer II/III 0.3 ± 0.07, layer V/VI 0.3 ± 0.1, $P = 0.975$; 24 Hz: layer II/III 0.16 ± 0.04, layer V/VI 0.26 ± 0.11, $P = 0.399$; 28 Hz: layer II/III 0.08 ± 0.03, layer V/VI 0.24 ± 0.11, $P = 0.192$; 32 Hz: layer II/III 0.14 ± 0.05, layer V/VI 0.2 ± 0.09, $P = 0.561$; 64 Hz: layer II/III 0.02 ± 0.01, layer V/VI 0.04 ± 0.01, $P = 0.18$) (Figs 2f and 3f). These data show that only

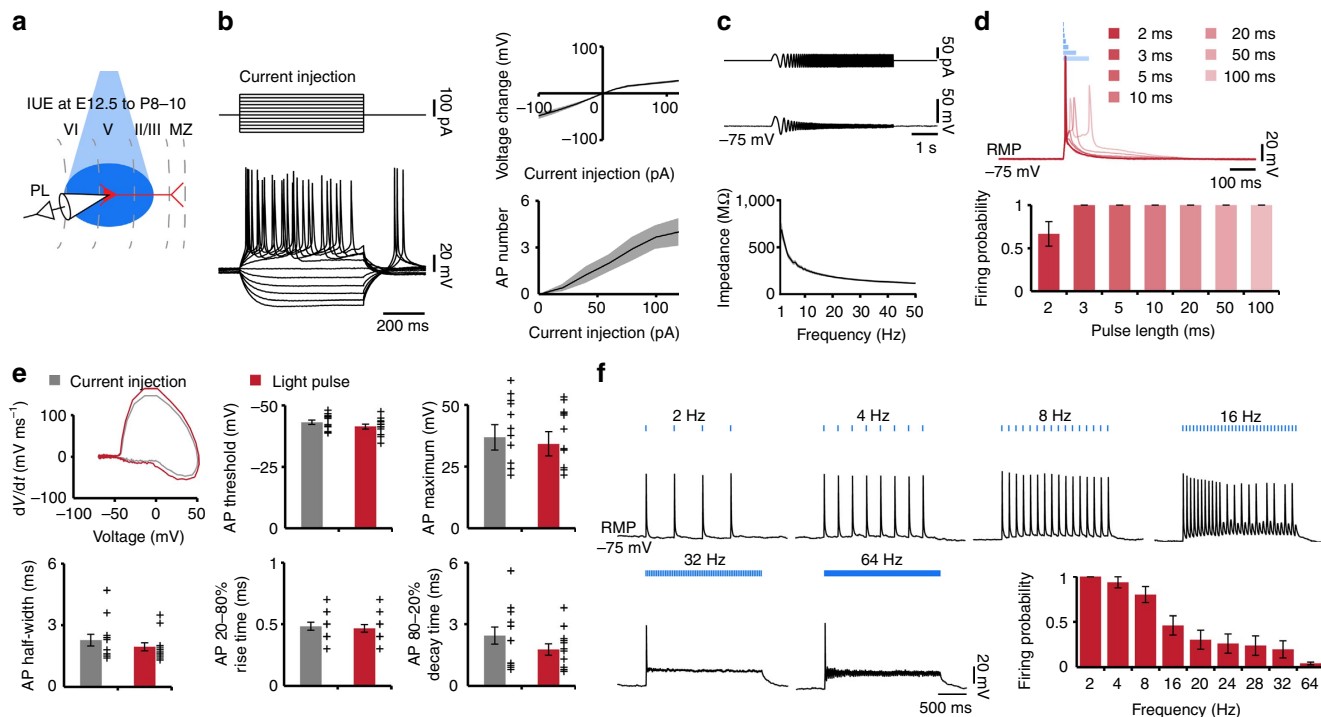

**Figure 3 | Optogenetic activation of prelimbic layer V/VI pyramidal neurons *in vitro*.** (**a**) Experimental approach for combined light stimulation and whole-cell recordings from IUE-transfected layer V/VI pyramidal neurons in coronal slices of P8–10 mice. (**b**) Left, voltage responses of a representative neuron to hyper- and depolarizing current pulses. The RMP was −68.6 mV. Right, averaged plots of the current–voltage relationship (top) and current-firing rate relationship (bottom) (*n* = 12 neurons). (**c**) Top, waveform of subthreshold chirp current injection with increasing frequency from 1 to 50 Hz. Middle, voltage response of a representative neuron. Bottom, averaged plots of the impedance (*n* = 12 neurons). (**d**) Top, voltage responses of a representative ChR2(ET/TC)-transfected neuron to blue light pulses (473 nm, 5.2 mW mm$^{-2}$) of 1 (dark red) to 100 ms (light red) duration. Bottom, bar diagram displaying the mean firing probabilty of transfected neurons in response to blue light pulses of variable duration (*n* = 12 neurons). (**e**) Top, representative phase plot of APs elicited in a transfected neuron by depolarizing current injection (grey) or a 3-ms-long light pulse (473 nm, 5.2 mW mm$^{-2}$, red). Bottom, bar diagrams displaying the mean active membrane properties after depolarizing current injections (grey, *n* = 12 neurons) and after light stimulation (red, *n* = 12 neurons). Individual values are displayed as black crosses. (**f**) Representative voltage responses of a transfected neuron to repetitive trains of 3-ms-long light stimuli at variable frequencies. Bar diagram displaying the mean firing probability of transfected neurons in response to repetitive light stimuli (*n* = 12 neurons). Data are presented as mean ± s.e.m. Two-sided *t*-tests. MZ, marginal zone.

for 16 Hz the firing probability was significantly higher for layer II/III when compared to layer V/VI pyramidal neurons. At stimulation frequencies > 32 Hz the firing probability strongly decreased and the few APs triggered at the beginning of the stimulation train were followed by a prominent plateau depolarization. These values are consistent with the maximal firing rate (layer II/III 10.0 ± 0.4 Hz; layer V/VI 8.0 ± 0.4 Hz) of these pyramidal neurons in response to depolarizing current injection.

In addition to identifying intrinsic differences in the activation of layer II/III and V/VI pyramidal neurons by light, the results of *in vitro* experiments were instrumental for setting light stimulation *in vivo*. All further manipulation of neonatal ChR2(ET/TC)-expressing pyramidal neurons were performed with 3-ms-long light pulses at frequencies up to 32 Hz.

**Layer II/III pyramidal neurons tend to fire in beta-rhythm.** To determine whether layer II/III and layer V/VI pyramidal neurons have distinct functions for the emergence of prelimbic network activity at neonatal age, we monitored the effects of their light activation *in vivo*. If these neurons contribute to the generation of oscillatory activity in the developing PL, then light stimulation should cause a concentration of their firing within a specific frequency band. Multisite recordings of the LFP and multiunit

activity (MUA) were performed from layer II/III or layer V/VI of the PL in urethane-anesthetized P8–10 mouse pups before, during and after repetitive stimulation with pulse trains or ramp light stimuli (Supplementary Fig. 3a). Our previous data revealed that network oscillations and neuronal firing are similar in urethane-anesthetized and asleep non-anesthetized neonatal rodents[8]. For each pup, the intensity of light stimulation was set to evoke reliable MUA activity < 15 ms after stimulus onset (Supplementary Fig. 3b,c) and ranged between 20 and 40 mW mm$^{-2}$. Stimulation efficacy increased with light power, yet the onset of evoked spiking remained constant (Supplementary Fig. 3d). To exclude that light-evoked effects are due to local tissue heating during illumination, we estimated the temperature change during stimulation using a recently developed model[40]. Stimulation with the illumination parameters set *in vitro* (3-ms-long light pulses, frequencies ranging from 2 to 32 Hz) led to a temperature increase of max. 0.2 °C (Supplementary Fig. 3e–h). These values are below those that have been reported to augment neuronal firing[40,41], indicating that the neonatal brain activity is not affected by light-induced tissue heating.

Trains of pulse stimuli (3-ms-long pulses at 2–32 Hz, total duration 3 s) and ramp light stimuli (total duration 3 s) increased the neuronal firing of ChR2(ET/TC)-transfected pyramidal neurons in both upper and deeper layers, but not of neurons tranfected with opsin-free constructs (Fig. 4a,d, Supplementary

Fig. 4 and Supplementary Fig. 5). The effects were more prominent in layer II/III than in layer V/VI neurons. Similar to the *in vitro* results, the firing of deeper layer pyramidal neurons was not reliably detected after trains of light pulses. In the upper layers, MUA discharge and the firing of 34 out of 69 single units (SUA) augmented in a frequency-dependent manner when trains of stimuli were applied. Although for low stimulation frequencies (2–16 Hz), the spiking increase was similar after each light pulse, at higher frequencies (16–32 Hz) the stimulation efficacy strongly decreased during the train, confirming the *in vitro* findings (Supplementary Fig. 4b–f). The onset of light-evoked spiking

(6.2 ± 0.5 ms, $n = 29$ recording sites from 10 mice) was similar to the onset of APs recorded in prelimbic neurons in slices (7.9 ± 0.4 ms, $n = 14$ cells). To exclude the possibility that the layer-specific firing patterns induced by light stimulation resulted from activation of a different number of neurons, we assessed the number of activated neurons by using the model of spatiotemporal propagation of light[40]. Taking into account the density of transfected and all neurons (layer II/III 3,698.8 ± 108.7 NeuN$^+$ per mm², layer V/VI 3,245.2 ± 121 NeuN$^+$ per mm², $P = 0.007$), similar numbers of neurons were estimated to be activated (illumination $> 10\,\text{mW mm}^{-2}$) in layer II/III (363 neurons) and layer V/VI (303 neurons). This suggests a comparable light activation for both layers; however, a contribution of minor differences in the activated number of neurons for superficial and deep layers to the observed differences remains possible.

Ramp stimulation of transfected neurons in upper and deeper layers led to sustained increase of spike discharge that was initiated once the power exceeded a certain threshold (Fig. 4a,d). For some neurons the activity dropped towards the end of ramp stimulation, suggesting that, similar to *in vitro* recordings, their membrane potential reached a depolarized plateau potential preventing further spiking. However, for the majority of neurons the mean firing rate for 1 s after stimulus remained significantly higher than before stimulus (layer II/III—before: 1.0 ± 0.1 Hz, after: 2.9 ± 0.5 Hz, $P = 0.048$, $n = 21$ recording sites; layer V/VI—before: 1.1 ± 0.1 Hz, after: 3.9 ± 0.3 Hz, $P < 0.001$, $n = 61$ recording sites), suggesting that short-term plasticity has been induced by light stimulation in developing circuits. Ramp stimulation revealed major differences between the firing of upper and deeper layers. While layer II/III neurons did not fire randomly, but had a preferred interspike interval of ∼60 ms, equivalent to a concentration of individual and population firing at 16.7 Hz (Fig. 4b,c), layer V/VI neurons lack this coordinated firing (Fig. 4e,f). The beta band concentration of firing in upper prelimbic layers was detected solely during ramp stimulus and was absent before and afterwards. The absence of a preferred membrane resonance of layer II/III pyramidal neurons (Fig. 2c) suggests that this firing pattern is not intrinsic, but emerges from cellular interactions within local networks. In line with the persistent poststimulus augmentation of firing rates, the proportion of short interspike intervals was larger after ramp light stimulus than before (Fig. 4b,c,e,f).

These results indicate that on light activation pyramidal neurons in layer II/III but not layer V/VI of neonatal PL concentrate their firing in beta frequency range.

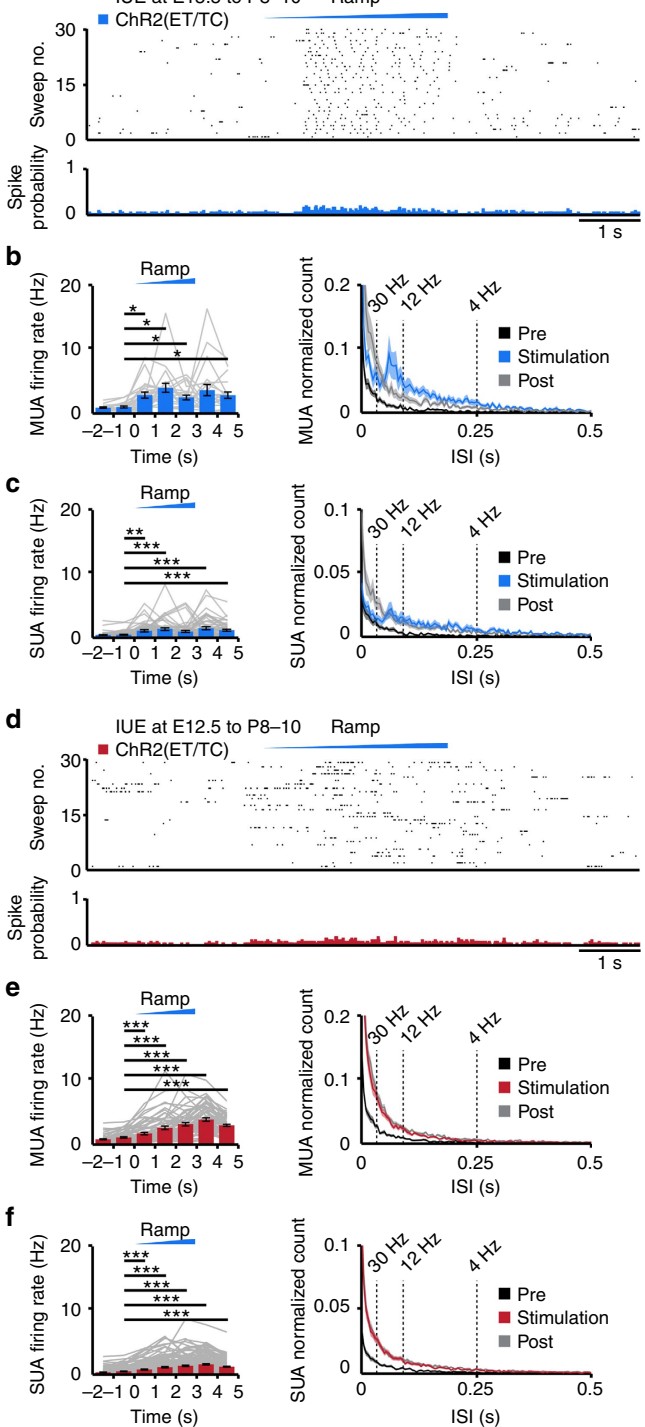

**Figure 4 | Optogenetic activation of layer II/III and layer V/VI pyramidal neurons *in vivo*.** (**a**) Representative raster plot and corresponding spike probability histogram displaying the firing of transfected layer II/III pyramidal neurons in response to 30 sweeps of ramp illumination (473 nm, 3 s). (**b**) Left, bar diagram displaying the mean MUA frequency in ChR2(ET/TC)-transfected neurons from P8 to P10 mice in response to ramp illumination. Right, occurrence rate of interspike intervals averaged for 3 s before light stimulation (pre, black), 3 s during ramp stimulation (stimulation, blue) and 3 s after light stimulation (post, grey, $n = 21$ recording sites from 9 pups). Note the prominent increase in the rate of interspike intervals between 12 and 30 Hz peaking at 16.7 Hz during ramp stimulation. (**c**) Same as (**b**) for SUA ($n = 50$ units from 9 pups). Note that the peak between 12 and 30 Hz is present, but less pronounced due to the incorporation of non-triggered units. (**d–f**) Same as (a–c) for layer V/VI pyramidal neurons (red, $n = 12$ pups; MUA of $n = 61$ recording sites; SUA of $n = 152$ units). For (**b,c,e,f**) grey lines correspond to firing of individual neurons. Data are presented as mean ± s.e.m. \*$P < 0.05$, \*\*$P < 0.01$ and \*\*\*$P < 0.001$), one-way repeated-measures analysis of variance (ANOVA) with Bonferoni-corrected *post hoc* analysis.

**Layer II/III pyramidal neurons drive beta–gamma oscillations.** Activation of pyramidal neurons in upper layers has been proposed to underlie the emergence of network oscillations in beta–gamma frequency range in the neonatal PL[8]. To causally prove this hypothesis, we tested whether the frequency-specific synchronization of layer II/III pyramidal neurons on light stimulation boosts the generation of discontinuous oscillatory activity in P8–10 mice. As previously reported[7,34], the first patterns of network activity in the neonatal PL are discontinuous, that is, spindle-shaped oscillations switching between theta and beta–gamma frequency components alternated with long periods of silence (Supplementary Fig. 6a,d). Selective expression of ChR2(ET/TC) in pyramidal neurons of prelimbic upper or deeper layers did not perturb the spontaneous discontinuous oscillatory events. In particular, their duration, amplitude, occurrence and spectral composition as well as their temporal relationship with the neuronal firing as quantified by phase locking were similar in pups transfected with ChR2(ET/TC)-containing constructs (layer II/III $n = 28$ pups, layer V/VI $n = 17$ pups) and pups transfected with opsin-free constructs (layer II/III $n = 11$ pups, layer V/VI $n = 12$ pups) (Supplementary Fig. 6b,c,e,f).

In a first step, we used trains of light pulses (3-ms-long, total duration of 3 s) to drive the spiking of either layer II/III or layer V/VI neurons in the PL of mice transfected with ChR2(ET/TC) at different frequencies (2, 4, 8, 16 and 32 Hz) (Supplementary Fig. 7a,b). Simultaneously, we recorded the LFP in PL. As previously reported, scattered photons hitting the recording sites of LFP electrodes led to prominent light artefacts[42]. They were measured at the end of the recordings in pups that were killed (see Methods) and eliminated by subtraction after corresponding scaling (Supplementary Fig. 7c,d). In opsin-free pups the procedure led to abolishment of the light-induced response, which consisted only of the photoelectric artefact. In contrast, in ChR2(ET/TC)-transfected pups, large negative voltage deflections with slower kinetics persisted after elimination of the photoelectric artefact. They seem to reflect physiological current sinks created in the extracellular space by simultaneous opening of the light-activated channels[43]. These large light-triggered responses precluded the assessment of induced oscillations when analysing the LFP, since due to their periodicity they are prominently reflected in the LFP power (Supplementary Fig. 7a,c). Therefore, we compared the power of network oscillations in the upper PL layers before and after, but not during each train of light pulses that activates layer II/III pyramidal neurons. Stimulation at 2, 4 as well as 32 Hz did not significantly modify the LFP. In contrast, 8 and 16 Hz stimulation augmented the power of oscillatory activity in theta (4–12 Hz) (post/pre theta: 2 Hz: $1.20 \pm 0.15$ $P = 0.228$; 4 Hz: $1.02 \pm 0.12$ $P = 0.860$; 8 Hz: $1.32 \pm 0.12$ $P = 0.021$; 16 Hz: $1.31 \pm 0.08$ $P = 0.004$; 32 Hz: $1.17 \pm 0.17$ $P = 0.333$), beta (12–30 Hz) (post/pre beta: 2 Hz: $1.27 \pm 0.13$ $P = 0.069$; 4 Hz: $1.10 \pm 0.08$ $P = 0.265$; 8 Hz: $1.23 \pm 0.07$ $P = 0.012$; 16 Hz: $1.30 \pm 0.06$ $P = 0.001$; 32 Hz: $1.21 \pm 0.15$ $P = 0.202$) and gamma (30–100 Hz) (post/pre gamma: 2 Hz: $1.11 \pm 0.08$ $P = 0.189$; 4 Hz: $1.04 \pm 0.04$ $P = 0.354$; 8 Hz: $1.11 \pm 0.04$ $P = 0.030$; 16 Hz: $1.17 \pm 0.05$ $P = 0.006$; 32 Hz: $1.05 \pm 0.06$ $P = 0.385$) frequency bands (Supplementary Fig. 7b). The strongest effects were observed when layer II/III pyramidal neurons were driven at 16 Hz. To examine the time course of network activation during stimulation, we applied ramp stimulations. When compared with pulse train stimulations, they had the advantage of not inducing power contamination by repetitive and fast large-amplitude voltage deflections (Figs 5a and 6a) and to trigger more physiological and not artificially synchronous firing patterns. The LFP power in beta frequency component significantly increased on ramp stimulation of layer

II/III pyramidal neurons, whereas the prominent theta component remained unaffected during stimulation (Fig. 5b). At higher light intensity gamma band activity was also boosted. Even after stimulus, the augmented network activation persisted, yet lacked frequency specificity. To assess the influence of light-induced network activity on prelimbic firing, we calculated the phase locking of MUA and clustered SUA to oscillations. The firing of layer II/III neurons was more strongly phase-locked to beta (SUA: baseline: $0.18 \pm 0.01$; ramp: $0.26 \pm 0.03$; $P = 0.016$) and gamma (SUA: baseline: $0.45 \pm 0.02$; ramp: $0.58 \pm 0.03$; $P = 0.003$) activity during ramp stimulation when compared to their timing during spontaneous activity (Fig. 5c,d).

In contrast, light activation of layer V/VI pyramidal neurons did not cause frequency-specific boosting of oscillatory activity (Fig. 6a,b). Ramp stimulation of deep layers led to an overall power increase in theta, beta and gamma band that outlasted the stimulus. Consequently, no significant changes of phase-locking of neuronal firing to network oscillations were detected during or after light stimulation of layer V/VI pyramidal neurons (Fig. 6c,d). These layer-specific differences do not result from variant expression of light-sensitive proteins across pups. Two pieces of evidence support this conclusion. First, the fraction of pups with extraprelimbic transfection was similar after IUE at E12.5 and E15.5. Second, no significant correlations between expression strength and the firing rate or oscillatory power were detected (Supplementary Fig. 8).

To examine whether layer-specific stimulation by light differentially entrains the entire PL in oscillatory rhythms of defined frequency, we performed LFP recordings using 4-shank optoelectrodes covering the whole cortical depth. Light stimulation of layer II/III neurons ($n = 11$ pups) augmented the coherence and power, especially in beta and gamma band, within and between layers (Fig. 7a and Supplementary Fig. 9). In contrast, stimulation of layer V/VI pyramidal neurons ($n = 6$ pups) did not change the synchrony strength during stimulation and was followed by intralayer coupling in beta–gamma range poststimulus (Fig. 7b). Interestingly, no significant changes in theta-band coherence were observed after light stimulation of upper and deeper layers, despite increased theta-band power in the poststimulation period. This suggests that external input, most likely hippocampal drive[7,34,44], is required for theta-band activation of the PL.

These data identify pyramidal neurons in layer II/III of PL as a cellular substrate of beta–gamma network oscillations capable of synchronizing intra- and interlayer the neonatal PL, whereas layer V/VI pyramidal neurons contribute to the overall activation within developing networks (Fig. 8).

## Discussion

Combining selective optogenetic activation with extracellular recordings from neonatal mice *in vivo*, we provide causal evidence that cortical oscillations during early development can emerge as result of cell-type-specific activation within local intracortical circuits. Optical stimulation of ChR-expressing pyramidal neurons in layer II/III but not layer V/VI of the PL coordinated the neuronal firing and specifically boosted the emergence of discontinuous oscillatory activity in beta–gamma frequency bands.

Despite limited behavioural abilities, the developing cortex shows complex patterns of discontinuous network activity[45,46]. Their diversity (for example, fast discharge interspersing slow rhythms) and large frequency spectrum (that is, from delta band to high-frequency oscillations) even at such early age lead to the question whether these early network oscillations control the functional maturation of the neocortex. One prerequisite for

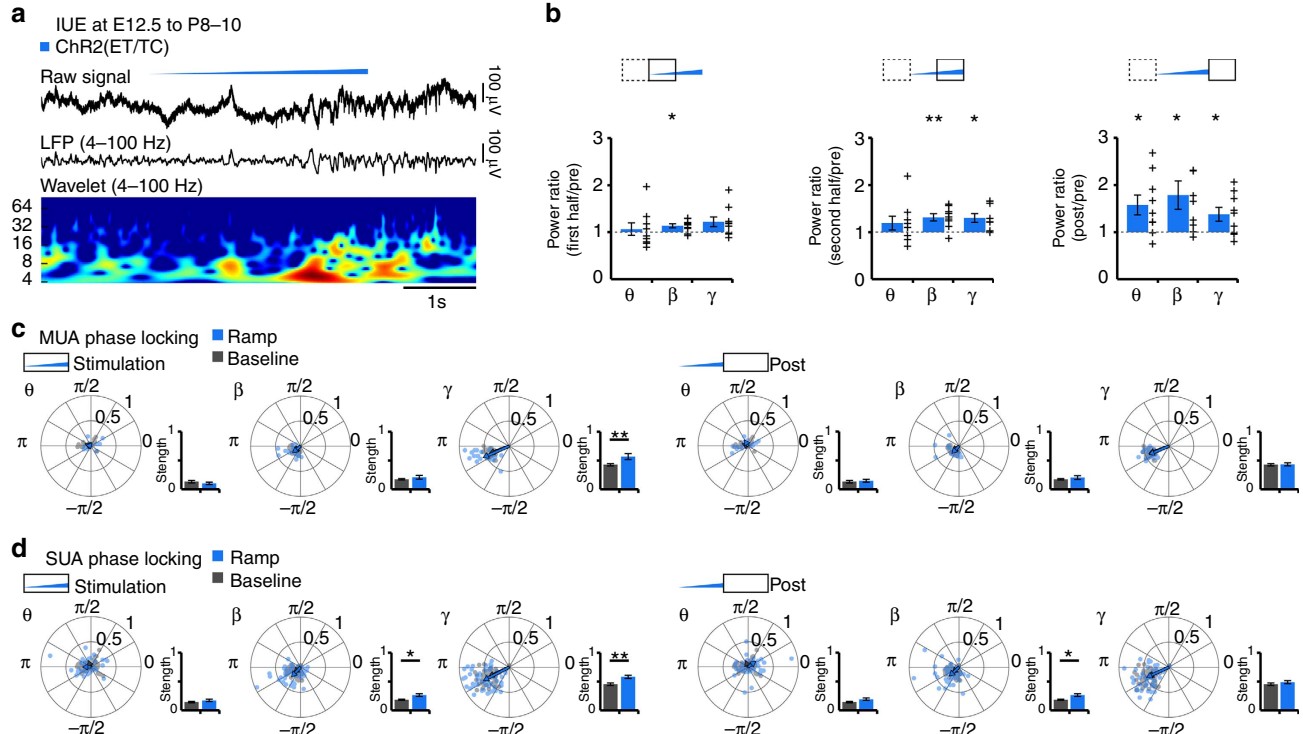

**Figure 5 | Generation of discontinuous patterns of oscillatory activity in the neonatal prelimbic cortex in response to optogenetic activation of layer II/III pyramidal neurons *in vivo*.** (**a**) Characteristic light-induced (ramp stimulus, 473 nm, 3 s) discontinuous oscillatory activity from a P10 mouse after transfection of layer II/III pyramidal neurons with ChR2(ET/TC) by IUE. The LFP is displayed before (top) and after band-pass filtering (4–100 Hz) (middle) together with the corresponding colour-coded wavelet spectrum at identical timescale. (**b**) Bar diagrams displaying the LFP power during the first half (1.5 s), the second half (1.5 s) and after (post, 1.5 s) ramp stimulus when normalized to the power before stimulation (pre, 1.5 s). Network activity in theta (θ, 4–12 Hz), beta (β, 12–30 Hz) and gamma (γ, 30–100 Hz) frequency bands ($n = 9$ pups) was considered. (**c**) Polar plots displaying the phase-locking of light-triggered (blue, stimulation 3 s, post 3 s) and spontaneous (grey) MUA to oscillatory activity from layer II/III ChR2(ET/TC)-transfected mice ($n = 21$ recording sites from 9 pups). Bar diagrams display the locking strength. (**d**) Same as (**c**) for SUA ($n = 50$ units from 9 pups). For (**b**) individual values corresponding to pups with light-induced oscillations are displayed as black crosses. For (**c**,**d**) the values from individual units are shown as blue and grey dots, whereas the arrows correspond to the mean resulting group vectors. Data are presented as mean ± s.e.m. $*P < 0.05$, $**P < 0.01$, two-sided $t$-tests and circular statistics toolbox.

elucidating the function of early oscillations is to understand their mechanisms of generation. Electrical stimulation and *in vivo* pharmacology showed that the early cortical activity is, at least in part, triggered by endogeneous activation of sensory periphery or by theta drive from other cortical and subcortical areas[2,3,7,25,47–49]. However, the partial persistence of early network oscillations after removal or blockade of these extraneocortical sources indicates that local activation of the neocortical circuitry may be sufficient for their generation. Indeed, we recently gained the first insights into the wiring scheme of layer V and showed that external glutamatergic inputs correlate with theta and beta–gamma activity[8]. However, none of these studies could precisely assign a specific neuronal population as generator of early patterns of network oscillations. The present study fills this knowledge gap by taking advantage of the selectivity of optogenetic manipulation and identifies one possible source of neonatal network activity. Using area-, layer- and cell-type-specific expression of a high-efficiency ChR mutant, we provide causal evidence that layer II/III pyramidal neurons, preferentially driven by light to fire at ~16 Hz, can entrain beta-band oscillatory activity in the neonatal PL. In contrast, activation of layer V/VI pyramidal neurons contributes to the overall oscillatory activation of neonatal prelimbic networks. When simultaneously monitored by 4-shank recordings, light activation of layer II/III and V/VI caused distinct synchrony patterns over PL. These data provide evidence for the existence

of different wiring schemes in layer II/III and V/VI. Their detailed elucidation will require simultaneous optogenetic targeting of both pyramidal neurons and interneurons in upper and deeper layers of PL. Another aspect that needs clarification in the future is to what extent the emerging mechanisms of early network oscillations are common for limbic and sensory cortices, which at adulthood have distinct structure, connectivity and behavioural relevance (for example, five versus six layers, prominent hippocampal versus thalamic innervation)[50,51].

Several considerations regarding the technical challenges of optogenetic manipulation of neonatal networks need to be made. First, the combination of the CAG promoter and targeting by IUE ensured a sufficiently high level of opsin expression in the neurons and layers of interest. Moreover, the number of transfected pyramidal neurons in the investigated layer II/III was sufficient to specifically induce coordinated fast network oscillations when stimulated with light. One intriguing question is how many pyramidal neurons in these layers must be synchronously activated to affect the network entrainment of the neonatal PL. Further technological development, such as high-density silicon probes combined with micron-sized light-emitting diodes could enable us to address this question in the future[52,53]. Second, the pattern of light stimulation appears critical for dissecting the cellular substrate of early network oscillations. Here we used both trains of pulses at different frequencies and ramp stimulation. Even after light artefacts were

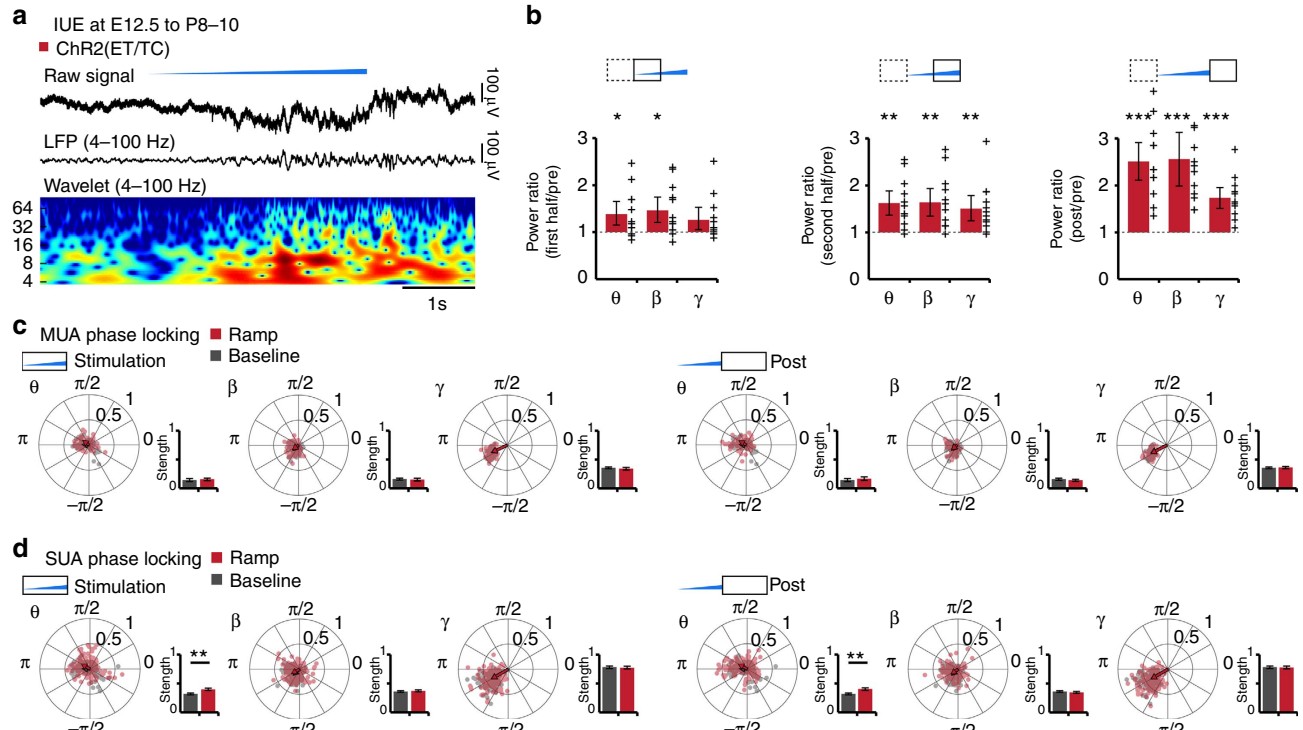

**Figure 6 | Generation of discontinuous patterns of oscillatory activity in the neonatal prelimbic cortex in response to optogenetic activation of layer V/VI pyramidal neurons *in vivo*.** (**a**) Characteristic light-induced (ramp stimulus, 473 nm, 3 s) discontinuous oscillatory activity from a P10 mouse after transfection of layer V/VI pyramidal neurons with ChR2(ET/TC) by IUE. The LFP is displayed before (top) and after band-pass filtering (4–100 Hz) (middle) together with the corresponding colour-coded wavelet spectrum at identical timescale. (**b**) Bar diagrams displaying the LFP power during the first half (1.5 s), the second half (1.5 s) and after (post, 1.5 s) ramp stimulus when normalized to the power before stimulation (pre, 1.5 s). Network activity in theta (θ, 4–12 Hz), beta (β, 12–30 Hz) and gamma (γ, 30–100 Hz) frequency bands (n = 12 pups) was considered. (**c**) Polar plots displaying the phase-locking of light-triggered (red, stimulation 3 s, post 3 s) and spontaneous (grey) MUA to oscillatory activity from layer V/VI ChR2(ET/TC)-transfected mice (n = 61 recording sites from 12 pups). Bar diagrams display the locking strength. (**d**) Same as (**c**) for SUA (n = 152 units from 12 pups). For (**b**) individual values corresponding to pups with light-induced oscillations are displayed as black crosses. For (**c,d**) the values from individual units are shown as red and grey dots, whereas the arrows correspond to the mean resulting group vectors. Data are presented as mean ± s.e.m. \*P < 0.05, \*\*P < 0.01, and \*\*\*P < 0.001, two-sided t-tests and circular statistics toolbox.

efficiently eliminated by subtraction[54], stimuli trains caused pronounced current sinks due to simultaneous opening of light-gated channels (large negative deflections of LFP in Supplementary Fig. 7c,d) that masked the induced network activity. This was not the case for ramp stimulation. Neuronal firing and early network oscillations were induced after a critical level of light intensity was reached. The LFP power gradually increased, persisting even after the stimulus. Similar to the activation of the adult brain[55,56], both train and ramp stimulations led to plastic boosting of activation within neuronal networks, as shown by the effects on LFP and MUA outlasting the stimulus. Its consequences for the functional maturation of the cortical circuitry remain to be assessed. Third, the entire investigation was performed on anesthetized neonatal mice. However, we note that previous comparisons of coordinated activity patterns in urethane-anesthetized and -non-anesthetized sleeping rodents identified no differences at neonatal age[8], when rodents sleep 70% of the time[57]. Therefore, we hypothesize that layer II/III and layer V/VI pyramidal neurons of the PL play a similar role as described here in anesthetized animals for the emergence of network oscillations in non-anesthetized naturally sleeping pups, although this remains to be tested directly.

In adults, optogenetic stimulation has been utilized to establish causal links between cell-type-specific activation and specific behavioural performance. By these means, the behavioural readout of fast beta and gamma oscillations has been characterized with relationship to their cellular substrates[58–60]. Modelling work proposed a dual origin of beta oscillations: they emerge either as 'slow gamma' within feedforward inhibitory networks including parvalbumin-expressing inteneurons, which are absent during neonatal development, or from the interplay between interneurons firing in gamma range and pyramidal neurons firing at lower beta frequency[61,62]. During early development, beta oscillations seem to be generated within an intracortical circuit that involves the activation of pyramidal neurons in layer II/III (shown here), which project to interneurons in layer V and boost their local inhibitory action entraining cortical circuits in gamma frequency band[8]. Taking into account the limited behavioural abilities of pups, it has been hypothesized that the complex patterns of early network activity do not have a direct behavioural readout but rather regulate the network wiring and neonatal plasticity, which is mandatory for later cortical function[8,63,64]. The present experimental strategy and findings open new perspectives for directly testing this hypothesis. Chronic optogenetic manipulation during defined developmental time windows (that is, critical/sensitive periods) of cellular elements identified as being necessary for the generation of early network oscillations will reveal, how the early cellular interplay controls circuit function later in life.

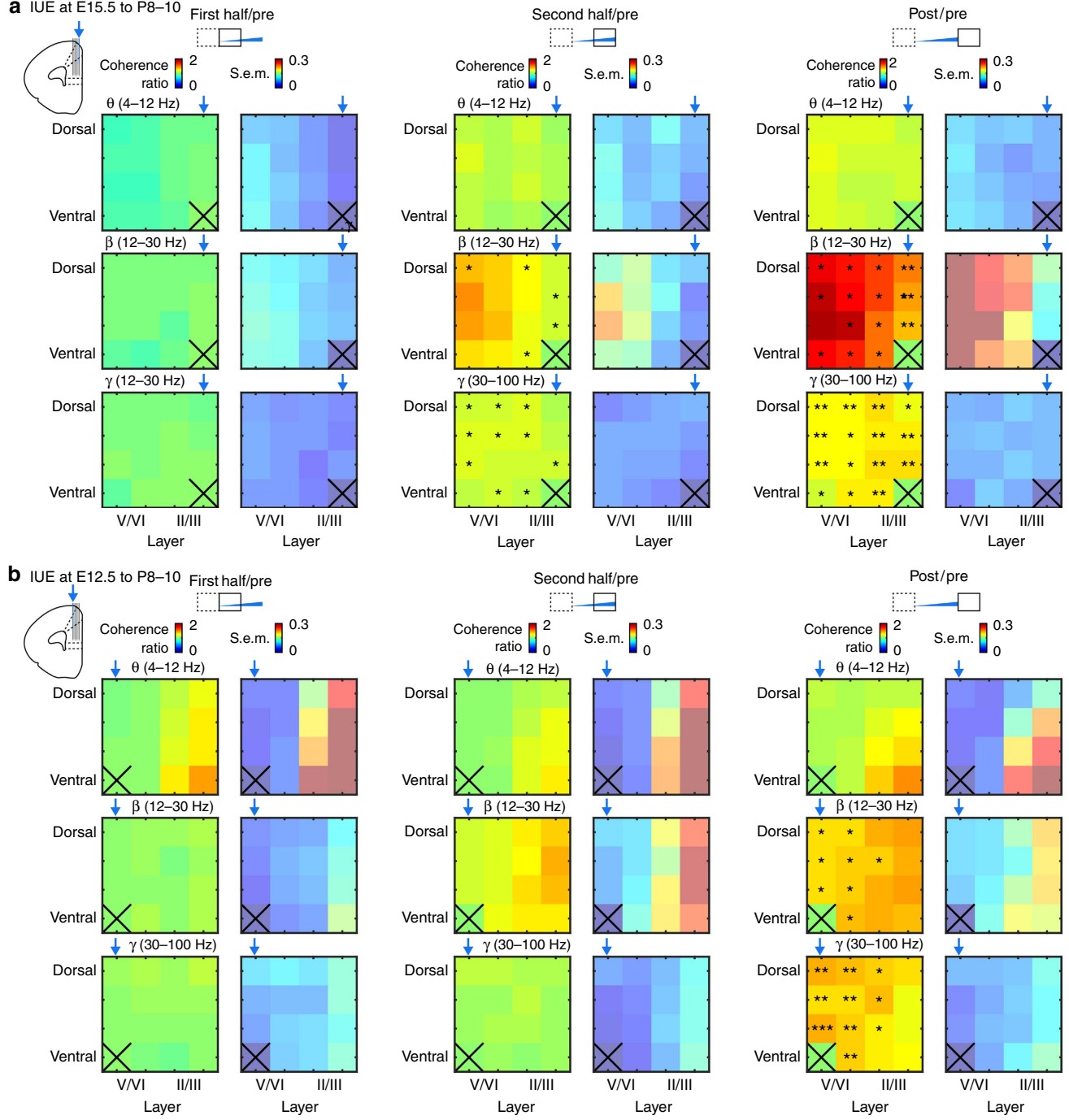

**Figure 7 | Frequency-dependent entrainment of neonatal prelimbic cortex in response to layer-specific optogenetic activation in vivo.** (**a**) Colour-coded images displaying the baseline normalized (pre, 1.5 s) coherence (mean and s.e.m.) between light-stimulated reference recording site in layer II/III (marked by X) and all other sites covering the PL depth during the first half (1.5 s), the second half (1.5 s) and after (post, 1.5 s) ramp stimulus. The coherence was calculated for theta (θ, 4–12 Hz, top), beta (β, 12–30 Hz, middle) and gamma (γ, 30–100 Hz, bottom) frequency bands and values were averaged ($n = 11$ pups). (**b**) Same as (**a**) for layer V/VI-expressing P8–10 mice ($n = 6$ pups). Reference recording site was located in the light-stimulated layer V/VI (marked by X). Blue arrows indicate the position of the light fibre. *$P < 0.05$, **$P < 0.01$ and ***$P < 0.001$, two-sided $t$-tests.

## Methods

**Animals.** All experiments were performed in compliance with the German laws and the guidelines of the European Community for the use of animals in research and were approved by the local ethical committee (111/12, 132/12). Experiments were performed on female and male C57Bl/6J mice at the age of P8–10 after IUE at E12.5 or E15.5.

**In utero electroporation.** Timed-pregnant C57Bl/6J mice from the animal facility of the University Medical Center Hamburg-Eppendorf were housed individually in

breeding cages at a 12 h light/12 h dark cycle and fed *ad libitum*. The day of vaginal plug detection was defined E0.5, while the day of birth was assigned as P0. Additional wet food was provided on a daily basis and was supplemented with 2–4 drops Metacam (0.5 mg ml$^{-1}$; Boehringer-Ingelheim, Germany) from one day before until two days after surgery. At E12.5 or E14.5–15.5 randomly assigned pregnant mice were injected subcutaneously with buprenorphine (0.05 mg/kg body weight) 30 min before surgery. The surgery was performed on a heating blanket and toe pinch and breathing were monitored throughout. Under isoflurane anesthesia (induction: 5%; maintenance: 3.5%), the eyes of the dam were covered with eye ointment to prevent damage before the uterine horns were exposed and

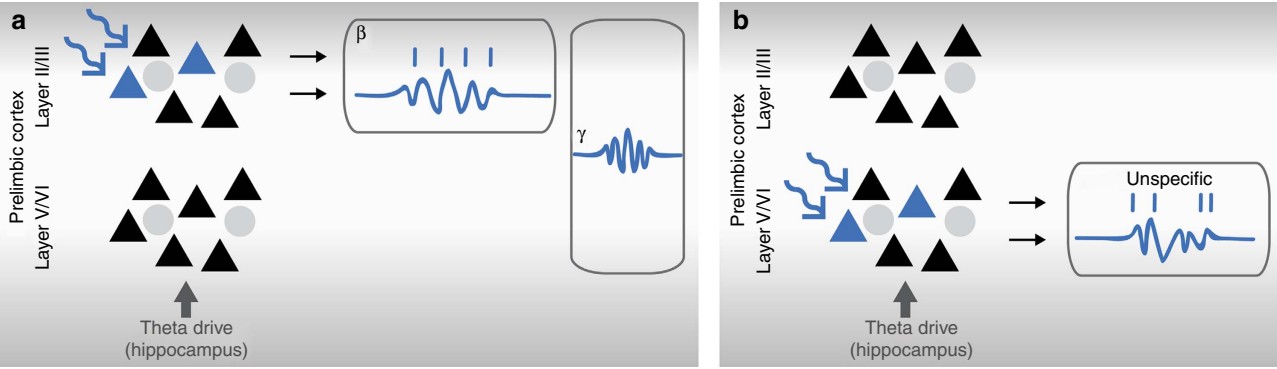

**Figure 8 | Schematic diagram depicting the contribution of layer II/III and layer V/VI to the generation of early oscillatory activity in distinct frequency bands investigated by optogenetic manipulation.** (**a**) Light stimulation of pyramidal neurons in layer II/III facilitates the generation of beta oscillations and increases the intra- (beta band) and interlayer (gamma band) synchrony. (**b**) Stimulation of pyramidal neurons in layer V/VI leads to network activation in all frequency bands and synchronizes solely deeper layers. Blue triangle, ChR2(ET/TC)-transfected pyramidal neurons; black triangle, non-transfected pyramidal neurons; grey circles, interneurons.

moistened with warm sterile PBS (37 °C). Solution containing 1.25 µg µl$^{-1}$ DNA (pAAV-EF1α-ChR2(E123T/T159C)-2A-tDimer2, pAAV-synapsin-ChR2(E123T/T159C)-2A-tDimer2, pAAV-CAG-ChR2(E123T/T159C)-2AtDimer2 or pAAV-CAG-tDimer2)) and 0.1% fast green dye at a volume of 0.75–1.25 µl were injected into the right lateral ventricle of individual embryos using pulled borosilicate glass capillaries with a sharp and long tip. Plasmid DNA was purified with NucleoBond (Macherey-Nagel, Germany). 2A encodes for a ribosomal skip sentence, splitting the fluorescent protein tDimer2 from the opsin during gene translation. Each embryo within the uterus was placed between the electroporation tweezer-type paddles (3 mm diameter for E12.5, 5 mm diameter for E14.5–15.5; Protech, TX, USA) that were oriented at a rough 20° leftward angle from the midline and a rough 10° angle downward from anterior to posterior. By these means, neural precursor cells from the subventricular zone, which radially migrate into the medial PFC, were transfected. Electrode pulses (35 V, 50 ms) were applied five times at intervals of 950 ms controlled by an electroporator (CU21EX; BEX, Japan). Most caudal embryos were not electroporated to minimize lethality. Uterine horns were placed back into the abdominal cavity after electroporation. The abdominal cavity was filled with warm sterile PBS (37 °C) and abdominal muscles and skin were sutured individually with absorbable and non-absorbable suture thread, respectively. The surgery was performed on a heating blanket, and toe pinch reflex and breathing were monitored. After recovery, pregnant mice were returned to their home cages, which were half placed on a heating blanket for two days after surgery. For most of the pups, opsin expression was assessed with a portable fluorescent flashlight (Nightsea, MA, USA) through the intact skull and skin at P2–3 and confirmed post mortem by fluorescence microscopy in brain slices. Pups without expression in the PFC were excluded from the analysis. Mice of both sexes were used.

**Behavioural examination.** Mouse pups were tested for their somatic development and reflexes at P2, P5 and P8. Weight, body and tail length were assessed. Surface righting reflex was quantified as time (max 30 s) until the pup turned over with all four feet on the ground after being placed on its back. Cliff aversion reflex was quantified as time (max 30 s) until the pup withdrew after snout and forepaws were positioned over an elevated edge. Vibrissa placing was rated positive if the pup turned its head after gently touching the whiskers with a toothpick.

**Histology and immunohistochemistry.** To quantify the transfection, P8–10 mice were anesthetized with 10% ketamine (aniMedica, Germany)/2% xylazine (WDT, Germany) in 0.9% NaCl solution (10 µg/g body weight, intraperitoneally (i.p.)) and transcardially perfused with Histofix (Carl Roth, Germany) containing 4% paraformaldehyde. Brains were postfixed in 4% paraformaldehyde for 24 h and sectioned coronally at 50 µm. Free-floating slices were permeabilized and blocked with PBS containing 0.8% Triton X-100 (Sigma-Aldrich, MO, USA), 5% normal bovine serum (Jackson Immuno Research, PA, USA) and 0.05% sodium azide. Subsequently, slices were incubated overnight with mouse monoclonal Alexa Fluor-488-conjugated antibody against NeuN (1:100, MAB377X; Merck Millipore, MA, USA), rabbit polyclonal primary antibody against CaM kinase II (1:200, PA5-38239; Thermo Fisher Scientific, MA, USA) or rabbit polyclonal primary antibody against GABA (1:1,000, no. A2052; Sigma-Aldrich), followed by 2 h incubation with Alexa Fluor-488 goat anti-rabbit IgG secondary antibody (1:500, A11008; Merck Millipore). Slices were transferred to glass slides and covered with Fluoromount (Sigma-Aldrich). Wide field fluorescence images were acquired to reconstruct the recording electrode position in brain slices of electrophysiologically investigated pups and to localize tDimer2 expression in pups after IUE. High

magnification images were acquired with a confocal microscope (DM IRBE, Leica, Germany) to quantify tDimer2 expression and immunopositive cells (4 brain slices per investigated mouse). All images were similarly analysed with ImageJ.

***In vitro* electrophysiology paired with optogenetic stimulation.** Whole-cell patch-clamp recordings were performed from fluorescently labelled layer II/III and layer V/VI prelimbic neurons in brain slices of P8–10 mice after IUE at E15.5 and E12.5, respectively. Pups were decapitated, brains were removed and immediately sectioned coronally at 300 µm in ice-cold oxygenated high sucrose-based artificial cerebral spinal fluid (ACSF) (in mM: 228 sucrose, 2.5 KCl, 1 NaH$_2$PO$_4$, 26.2 NaHCO$_3$, 11 glucose, 7 MgSO$_4$; 320 mOsm) with a vibratome. Slices were incubated in oxygenated ACSF (in mM: 119 NaCl, 2.5 KCl, 1 NaH$_2$PO$_4$, 26.2 NaHCO$_3$, 11 glucose, 1.3 MgSO$_4$; 320 mOsm) at 37 °C for 45 min before cooling to room temperature and superfused with oxygenated ACSF in the recording chamber. tDimer2-positive neurons were patched under optical control using pulled borosilicate glass capillaries (tip resistance of 4–7 MΩ) filled with pipette solution (in mM: 130 K-gluconate, 10 HEPES, 0.5 EGTA, 4 Mg-ATP, 0.3 Na-GTP, 8 NaCl; 285 mOsm, pH 7.4). Recordings were controlled with the Ephus software[65] in the Matlab environment (MathWorks, MA, USA). Capacitance artefacts and series resistance were minimized using the built-in circuitry of the patch-clamp amplifier (Axopatch 200B; Molecular devices, CA, USA). Responses of neurons to hyper- and depolarizing current injections, as well as blue light pulses (473 nm, 5.2 mW mm$^{-1}$) were digitized at 5 kHz in current-clamp mode. Linearly increasing chirp current injections were applied for membrane resonance measurements. Impedance was calculated as ratio of Fourier transforms of the measured voltage response to chirp current injections.

***In vivo* electrophysiology combined with optogenetic stimulation.** Multisite extracellular recordings were performed in the PL of P8–10 mice. Mice were injected i.p. with urethane (1 mg/g body weight; Sigma-Aldrich) before surgery. Under isoflurane anesthesia (induction: 5%; maintenance: 2.5%), the head of the pup was fixed into a stereotaxic apparatus using two plastic bars mounted on the nasal and occipital bones with dental cement. The bone above the PFC (0.5 mm anterior to bregma, 0.1 mm right to the midline for layer II/III, 0.5 mm for layer V/VI) was carefully removed by drilling a hole of <0.5 mm in diameter. After a 10–20 min recovery period on a heating blanket, one- or four-shank multisite optoelectrodes (NeuroNexus, MI, USA) were inserted 2–2.4 mm deep into PFC perpendicular to the skull surface. One-shank optoelectrodes contained 1 × 16 recordings sites (0.4–0.8 MΩ impedance, 100 µm spacing) aligned with an optical fibre (105 µm diameter) ending 200 µm above the top recording site. Four-shank optoelectrodes contained 4 × 4 recording sites (0.4–0.8 MΩ impedance, 100 µm spacing, 125 µm intershank spacing) aligned with optical fibres (50 µm diameter) ending 200 µm above the top recording sites. A silver wire was inserted into the cerebellum and served as ground and reference electrode. Extracellular signals were band-pass filtered (0.1–9,000 Hz) and digitized (32 kHz) with a multichannel extracellular amplifier (Digital Lynx SX; Neuralynx, Bozeman, MO, USA) and the Cheetah acquisition software (Neuralynx). Spontaneous (that is, not induced by light stimulation) activity was recorded for 15 min at the beginning and end of each recording session as baseline activity. Pulsed (laser on-off) and ramp (linearly increasing power) light stimulations were performed with an arduino uno (Arduino, Italy) controlled diode laser (473 nm; Omicron, Austria). Laser power was adjusted to trigger neuronal spiking in response to >25% of 3-ms-long light pulses at 16 Hz. Resulting light power was in the range of 20–40 mW mm$^{-1}$ at the fibre tip. Light-induced LFP artefacts were measured at the end of the experiment.

For this, mice were killed with an injection of 10% ketamine/2% xylazine in 0.9% NaCl solution (20 µg/g body weight, i.p.) abolishing brain activity while maintaining the optoelectrode position. Stimulation protocols were repeated 15 min after the lethal injection and the photoelectric artefacts were eliminated from alive recordings by subtraction after scaling to the immediate downstroke (that is, negative deflection 0–1.5 ms after the start of the light pulse) of the alive recordings in response to light pulses.

**Estimation of light and heat propagation.** The spatiotemporal propagation of light and heat were estimated using a recently developed model for *in vivo* data[40] for pulsed illumination (473 nm, 3 s, 3 ms pulses) at 2 to 32 Hz and 1 to 10 mW light power at the fibre tip (105 µm, numerical aperture 0.22) with optical absorption parameters for brain tissue[66]. Measured light power at the fibre tip before inserting the optoelectrode was in the range from 1 to 5 mW.

**AP Action potential modelling.** The conductance-based Hodgkin–Huxley model solved with the Euler method was used to investigate the influence of $Na^+/K^+$ conductance changes during development on AP properties in response to current injections[39]. APs were modelled with different levels of $Na^+/K^+$ conductance ($Na^+$: $0.6\,mS\,cm^{-2}$; $K^+$: $0.18\,mS\,cm^{-2}$ multiplied by 1.0 to 3.0) and AP properties in response to current injections were calculated.

**Data analysis.** Data were imported and analysed offline using custom-written tools in the Matlab software (MathWorks). For *in vitro* data, all potentials were corrected for liquid junction potentials with $-10\,mV$ for the gluconate-based electrode solution[67]. The RMP was measured immediately after obtaining the whole-cell configuration. For the determination of the input resistance, hyperpolarizing current pulses of 200 ms duration were applied. Active membrane properties and current–voltage relationships were assessed by unsupervised analysis of responses to a series of 600 ms long hyper- and depolarizing current pulses. Amplitude of APs was measured from threshold to peak.

*In vivo* data were processed as following: band-pass filtered (500–5,000 Hz) to analyse MUA and low-pass filtered (<1,500 Hz) using a third-order Butterworth filter before downsampling to 3.2 kHz to analyse LFP. All filtering procedures were performed in a manner preserving phase information. MUA was detected as the peak of negative deflections greater than five times the standard deviation of filtered signals. SUA was detected and clustered using Offline Sorter (Plexon, TC, USA) and 1–4 single units were detected at each recording site. Spikes occurring in a 15 ms time window after the start of a light pulse were considered to be light-evoked. Stimulation efficacy was calculated as the probability of at least one spike occurring in this period. Discontinuous network oscillations in the LFP were detected using a previously developed unsupervised algorithm[34]. Briefly, deflections of the root mean square of band-pass filtered signals (1–100 Hz) exceeding a variance-depending threshold were assigned as network oscillations. The threshold was determined by a Gaussian fit to the values ranging from 0 to the global maximum of the root-mean-square histogram. Only oscillations lasting >1 s were considered for further analysis. Time–frequency plots were calculated by transforming the data using Morlet continuous wavelet. For periods with light stimulations, photoelectric artefacts recorded post mortem were filtered (1–400 Hz), averaged, scaled to the immediate downstroke (0–1.5 ms after light onset) of the alive recordings and subtracted from the alive recordings. This immediate downstroke reflects the photoelectric artefact and is largely independent of currents sinks created by synchronized opening of ChRs. MUA band was not contaminated by light artefacts.

Statistical analyses were performed using SPSS Statistics 21 (IBM, NY, USA) or Matlab. Data were tested for normal distribution by the Shapiro–Wilk test. Normally distributed data were tested for significant differences (*$P<0.05$, **$P<0.01$ and ***$P<0.001$) using paired *t*-test, unpaired *t*-test or one-way repeated-measures analysis of variance with Bonferoni-corrected *post hoc* analysis. Not normally distributed data were tested with the nonparametric Mann–Whitney *U*-test. The circular statistics toolbox was used to test for significant differences in the phase locking data. Data are presented as mean ± s.e. of the mean. No statistical measures were used to estimate sample size since effect size was unknown. Investigators were not blinded to the group allocation during the experiments. Unsupervised analysis software was used if possible to preclude investigator biases.

**Data availability.** The authors declare that all data and code supporting the findings of this study are included in the manuscript and its Supplementary Information or are available from the corresponding authors on request.

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

## Acknowledgements

We thank Annette Marquardt, Iris Ohmert and Achim Dahlmann for technical assistance. This work was funded by grants from the European Research Council (ERC-2015-CoG 681577 to I.L.H.-O.) and the German Research Foundation (SPP 1665 to I.L.H.-O. and T.G.O., SFB 936 B5 to I.L.H.-O. and B7 to T.G.O., FOR 2419 P7 to J.S.W. and C.E.G., FOR 2419 P4 to T.G.O. and SPP 1926 to J.S.W.).

## Author contributions

I.L.H.-O. designed the experiments, S.H.B. and J.A. carried out the experiments, S.H.B. and J.A. analysed the data, A.W., C.E.G., J.S.W. and T.G.O. contributed to the establishment of experimental protocols and provided the constructs, I.L.H.-O., S.H.B. and J.A. interpreted the data and wrote the paper. All authors discussed and commented on the manuscript.

## Additional information

**Competing financial interests:** The authors declare no competing financial interests.

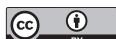

