## [Peer Review File · Nature Communications]

Reviewers' comments:

Reviewer #1 (Remarks to the Author):

A and B: Summary of the key results & Originality and interest: if not novel, please give references

The authors used in-utero electroporation to express a variant of ChR2 specifically in LII/III or LV/VI of the prelimbic cortex of neonatal mice. To my knowledge this is the first attempt combining in utero electroporation with optogenetics - so the technique is interesting by itself. By illuminating the transfected area and simultaneously recording from mouse pups in vitro and in vivo, they further showed differences in responses between the different layers, leading to the main finding of this manuscript, stating that "...light activation of layer II/III pyramidal neurons ... drives frequency-specific spiking and boosts discontinuous network oscillations within beta-gamma frequency range. In contrast, driving layer V/VI pyramidal neurons by light causes non-specific network activation...". If this finding turns out to be valid, it would move the field of maturation of neuronal circuits and neural oscillations substantially forward.

I'm very enthusiastic about this study and I definitely would like to see it published. I have a couple of comments and suggestions though which would strengthen the manuscript.

C, E, F, H: Data & methodology: validity of approach, quality of data, quality of presentation; Conclusions: robustness, validity, reliability; Suggested improvements: experiments, data for possible revision; Clarity and context: lucidity of abstract/summary, appropriateness of abstract, introduction and conclusions

The manuscript is well written and has an easy to follow flow.

Major concern: While I'm enthusiastic about this study, I have one major concern. The strategy used to achieve layer specificity by electroporation in different embryonic days, led to a difference in expression profile between the 2 conditions (fig 1c compared to fig 1f). As the authors state, the emergence of oscillations upon light stimulation is probably not due to intrinsic resonance properties of the pyramidal cells (fig. 2c for L2-3 neuron. Similar analysis for L5 neurons is missing). From that, the authors suggested reasonably that "...this firing pattern does not emerge intrinsically, but emerges from cellular interactions within local networks." (Line 240). However, by comparing fig. 3a to fig. 3d, it seems like light stimulation in L2-3 excites firing more efficiently and reliably than in L5-6. While the L2-3 neuron appears to be a part of a network, the L5-6 neuron seems to increase momentarily its firing rate upon reaching a threshold, and then to "saturate", which is suggestive for activation of a neuron not part of its network. In addition, while L2-3 units followed light pulse trains in different frequencies, L5-6 MUAs didn't (lines 219-224). Also, is there a reason why there is no SUA analysis for LV/VI?

While the conclusion of the authors that the differences between the different layers reflects a mechanistic difference in involvement in neonatal oscillations may indeed be valid, other interpretations are plausible given the current results. The most severe one is that ChR2 positive neurons of E15.5 IUE are focally expressed in PL, while those of E12.5 IUE have bigger distances between them and are scattered also to other cortical areas. This may lead to recruitment of a bigger (relative) part of a local network in the 15.5 IUE condition (assuming all experimental conditions, especially fiber diameter and distance from recording site, are the same). In other words- it might be that in the 15.5 IUE condition, a network was activated by light, while in the 12.5 IUE condition just a small fraction of it (or a single neuron in the extreme case) was activated.

Therefore, in order to validate the important message which the authors want to convey, a more comparable expression profile of the two conditions (layer II/III) versus layer V/VI would be more convincing. For this, couldn't one of the transgenic Thy1-ChR2 lines be used in order to achieve a higher specificity for targeting specific layers? From CAMPSALL et al (2002, Developmental

Dynamics) 'Characterization of Transgene Expression and Cre Recombinase Activity in a Panel of Thy-1 Promoter-Cre Transgenic Mice' it seems that some lines show already expression of an integrated gene as early as day P14. If this approach is not feasible this alternative should at least be discussed as well as the potential confound mentioned above (network versus small fraction activation).

Minor remarks:

Line 51 (abstract): "entrainment of neonatal prefrontal networks in fast rhythms critically relies on a local glutamatergic drive...": there is no evidence for that statement in the text. The rhythms might be dependent on di-synaptic GABAergic activation. This claim requires at least pharmacological manipulation of glutamate and/ or GABA receptors.

line 107: were other types of adeno viruses used, e.g., would the most commonly used serotypes 2 or 5 work?

line 112: it would be beneficial to include an example figure of the "bad" viral expression, at least in comparison to IUE (especially after devoting a whole paragraph to viruses). Also, a figure about the comparison of CAG and hSyn/Ef1a should be provided.

line 137: A positive staining for pyramidal cells, such as CaMKII, should be done.

Supp fig. 1: not all pups survive the IUE, including the opsin-free treatment, suggesting the treatment is stressful. Therefore, a behavioral comparison to "IUE free" pups would be beneficial.

Line 165: a short explanation for why this measure was taken would be useful (pyramidal cells are non-resonators, so this is another evidence suggesting these are pyramidal cells, or to study if oscillations may be due to intrinsic or network properties)

Line 174: there is a trend towards lower peak amplitude of ChR2 cells. As the number of cells is quite low (5), it will be beneficial to test more cells and exclude the hamper of a different spiking profile.

Line 187: for making such a statement testing more frequencies between 16 and 32 Hz would be necessary.

Fig. 2: was a similar analysis done for cells in L5-6? (the fig is only about L2-3). If so, are the results comparable?

Line 222: AP latency of 6 ms seems quite long for directly activated neurons. From my experience it should be around 3 ms.

Line 243: in order to claim for synchronization in beta, the ISI is not sufficient as the only measurement. It should be accompanied with a phase-lock-analysis. Therefore, this claim should appear later in the manuscript.

Line 508: it is not clear what "scaled to the immediate downstroke" means

Language and visibility:

I suggest to present the main findings more clearly in one summarizing figure. In the current manuscript version the main information is hidden in subpanels and very hard to extract.

Supp fig 5 c and f: the arrows are hardly visible (dark contour, for example, would help)

Line 316: "led" should be "lead"

D: Appropriate use of statistics and treatment of uncertainties

Statistics seem to be fine to me.

G: References: appropriate credit to previous work?

Yes.

Reviewer #2 (Remarks to the Author):

In this work, Bitzenhofer et al. show that optogenetic activation of layer II/III pyramidal neurons in neonatal mouse prefrontal cortex (PL) can induce network oscillations in beta-gamma frequency

range, characteristic patterns of network activity observed in developing cortical circuits. They expressed highly-efficient channelrhodopsin mutant (ChR2ET/TC) in PL neurons by in utero electroporation (IUE), delivered light by an optical fiber, and recorded local field potentials, multi-unit and single-unit activities by multi-site electrodes. They found that light activation caused layer II/III pyramidal neurons to fire preferentially at ~16 Hz and boosted beta-gamma band oscillatory activity in neonatal PL. Layer V/VI neurons neither showed such frequency-specific firing nor caused frequency-specific boosting of oscillatory activity. Based on these findings, they suggest that entrainment of neonatal prefrontal networks in fast rhythms relies on a local glutamatergic drive and that layer II/III pyramidal neurons can serve as one generator of the neonatal network activity.

Whether their finding that layer II/III but not layer V/VI pyramidal neurons are the key for generating beta-gamma oscillations during the neonatal period can be generalized to other cortical areas remains an open question. In addition, what is mechanistically different between layer II/III and layer V/VI circuits is not well addressed. However, optogenetic activity manipulation in neonatal mouse brain is technically challenging. This study therefore provides novel and valuable information to many researchers in the field.

Specific comments:

1. I have a concern about the area specificity of IUE-mediated gene expression. The authors show only one brain section each for E15.5 IUE and E12.5 IUE (Fig1). In Fig1c (E15.5 IUE), it appears that tDimer2 expression is restricted to the PL. In Fig1f (E12.5 IUE), the expression appears widespread. How about the expression in other areas of other brain sections? If channelrhodopsin is expressed in neurons of other areas and they send axons to the PL, light activation experiments (such as those shown in Fig3 and 4) may be affected because Ch-expressing axons can be easily activated. More specifically, it may be that light activation targeting layer II/III, but not layer V/VI, caused frequency-specific boosting of oscillatory activity because channelrhodopsin expression was restricted to PL in E15.5 IUE but it was widespread in E12.5 IUE. I would like to see the authors provide sufficient information to deny this possibility.
2. In Supplementary Fig. 1d, the rate of successful expression in PL is relatively low. I wonder how the authors chose pups with channelrhodopsin expression in PL for in vivo recording experiments. Did they confirm gene expression in PL by some method before recording, or did they perform recording in all pups then confirm gene expression afterwards? I couldn't find their procedure in the manuscript.
3. In Fig4b legend, does "n=9" mean the number of animals or recordings? If it is the number of animals, could the specific network activity (beta or gamma) be reliably induced in all recording sessions in each animal? If it is the number of recordings, is the data from one animal? This is a question of how reliably could the authors induce beta/gamma band network activity across recording sessions and across animals. In addition, does the extent of channelrhodopsin expression correlate with the success rate of inducing the specific network activity?
4. Page 9, lines 287-291: the authors state that "To examine how light-induced network oscillations spread over PL, we monitored,,,, Augmented beta-band activity was detected,,,,". Is the data for this statement presented? I think this is related to the authors' discussion that "During early development, beta oscillations seem to be generated,,, boost their local inhibitory action entraining cortical circuits in gamma frequency band" (page 11, lines 366-369) and important.
5. Page 4, lines 117-118: References are required for "our previous investigations".
6. Page 5, line 156: does "n=5" mean the number of animals? Slices? Neurons? I thought this was the number of neurons, but found "n=12 neurons" in Fig2b legend and was confused.
7. Page 6, line 198: "than" should be "then".

8. Page 9, lines 276-277: the authors state that "did not significantly modify the LFP", but there is no statistical information.

9. In Fig legends of Fig3, 4, SuppleFig4, SuppleFig5, SuppleFig6, the information about which test was used is missing.

10. In Fig4b and Fig4f, boxes with dotted lines show the period before light stimulation. Is this 1.5s or 3s? In Fig4cd and Fig4g, do boxes with solid lines show 3s? How about the boxes in SuppleFig6b? In SuppleFig6b legend, does "n=10" mean the number of recordings or animals?

Reviewer #3 (Remarks to the Author):

This study uses a combination of different methods to elucidate the cellular and network mechanisms underlying beta-gamma activity in the prefrontal prelimbic (PL) cortex of P8-P10 mice. The main result is shown in figure 4: layer 2/3 (Fig 4b), to a minor extent layer 5/6 (Fig 4f) pyramidal neurons boost beta-gamma, to some extent also theta network activity.

The authors already previously uncovered the cellular interactions contributing to theta and beta-gamma oscillations in P6-P8 mice using similar recording techniques and identified L5 pyramidal cells and GABAergic interneurons as critical components to generate this network activity (Bitzenhofer et al., 2015).

The important role of L2/3 in generating spontaneous theta- and beta/gamma oscillations in developing cerebral cortex (>P8) has been already documented previously (Gireesh and Plenz, 2008). This paper is not even cited. Therefore a statement as "we provide the first causal demonstration that cortical oscillations during early development can emerge as result of cell type-specific activation within local intracortical circuits" (line 308) is not true.

The ms contains many misleading or even false statements. I just give a few examples from the first pages.

- In title and text authors use the term "neonatal", but mice older than P8 were studied. Considering the rapid development and the general terminology in rodent development, the term neonatal should not be used.
- First lines of Introduction: "For long time the developing brain has been considered as the premature form of the adult one. .. This dogma has been profoundly challenged". As we can read in every text book on developmental neurobiology from the last 40-50 years, the developing brain is of course not a premature form of the adult brain. Many processes are unique for the early development. This "dogma" simply does not exist.
- Line 68: "the depolarizing but inhibitory action of GABA-A": This is a very selective and incorrect form of citing the literature of the last 20 years from many labs.
- Line 68: "the directed interactions ensuring both local and long range network coupling are exclusively present during early postnatal development." Of course local and long-range network coupling is strongly expressed in the adult brain.
- Line 71: "Recently, specific neuronal sub-populations and their transient connectivity have been identified as being critical for proper maturation of neuronal circuits". Subplate cells, as one or

maybe the most important population of transient cells in the cerebral cortex and their connectivity have been described by Shatz and others not so "recently" (e.g. McConnell et al., 1989; Luskin and Shatz, 1985).

The ms also contains meaningless statements, such as "neuronal circuits as pre-adapted template of future behavioral requirements" (line 40).

About 70-80% of this study describes the methods (Results page 4-8, Figs. 1-3). Only the last chapter in Results on page 8f and Fig. 4 is related to the main question of this study. Looking at Fig 4 as the main result of this study I find the conclusion of this paper not convincing.

In their previous paper (Bitzenhofer et al., 2015) the authors also studied awake animals. This study would be more valuable if recordings also would have been done in non-anesthetized animals.

The behavioral tests used in this study are certainly not related to the transfected brain region. Reflexes are driven by spinal cord and not prefrontal cortex. Therefore a statement as "To exclude non-specific effects of the transfection with ChR2" is misleading (line 146).

line 328: "The preset study ... identifies one generator of neonatal network activity" is misleading. The "generator" maybe the presynaptic network activating L2/3 neurons in PL, but this network has not been identified.

Last line abstract: "This approach enables the interrogation of developing circuits and their later behavioral readout." But behavioral readout has been studied in P2-P8 animals only (line 411).

The main result of this study maybe trivial, namely that L2/3 pyramidal cells tend to fire in beta/low gamma frequency more strongly than L5/6 pyramidal cells. Unfortunately this question is not clearly addressed (e.g. Fig. 2) despite the fact that previous studies support this hypothesis (e.g. Wespata et al., 2004; Gireesh and Plenz, 2008).

To get a better understanding of the network activity, analyses of simultaneous local field potential recordings in upper and lower layers in addition to SUA and MUA data would be most valuable. This is very easy to do and gives more information on the local network activity than some data shown in the paper.

I doubt that the increasing ramp stimulation used in Fig. 3 and 4 over several seconds is very physiological. Which physiological synaptic input would provide such an activation of pyramidal cells in PL?

Bitzenhofer SH, Sieben K, Siebert KD, Spehr M, Hanganu-Opatz IL (2015) Oscillatory Activity in Developing Prefrontal Networks Results from Theta-Gamma-Modulated Synaptic Inputs. *Cell Reports* 11: 486-497.

Gireesh ED, Plenz D (2008) Neuronal avalanches organize as nested theta- and beta/gamma-oscillations during development of cortical layer 2/3. *Proc Natl Acad Sci U S A* 105: 7576-7581.

Luskin MB, Shatz CJ (1985) Studies of the earliest generated cells of the cat's visual cortex: cogeneration of subplate and marginal zones. *J Neurosci* 5: 1062-1075.

McConnell SK, Ghosh A, Shatz CJ (1989) Subplate neurons pioneer the first axon pathway from

the cerebral cortex. *Science* 245: 978-982.

Wespatat V, Tennigkeit F, Singer W (2004) Phase sensitivity of synaptic modifications in oscillating cells of rat visual cortex. *J Neurosci* 24: 9067-9075.

Reviewer #1 (Remarks to the Author):

I'm very enthusiastic about this study and I definitely would like to see it published. I have a couple of comments and suggestions though which would strengthen the manuscript.

The manuscript is well written and has an easy to follow flow.

We thank the referee for the comments and suggestions that were most helpful. We hope that the large amount of data and analyses that were added to the revised manuscript strengthen the conclusions of the study.

Major concern: While I'm enthusiastic about this study, I have one major concern. The strategy used to achieve layer specificity by electroporation in different embryonic days, led to a difference in expression profile between the 2 conditions (fig 1c compared to fig 1f). As the authors state, the emergence of oscillations upon light stimulation is probably not due to intrinsic resonance properties of the pyramidal cells (fig. 2c for L2-3 neuron. Similar analysis for L5 neurons is missing). From that, the authors suggested reasonably that "...this firing pattern does not emerge intrinsically, but emerges from cellular interactions within local networks." (Line 240). However, by comparing fig. 3a to fig. 3d, it seems like light stimulation in L2-3 excites firing more efficiently and reliably than in L5-6. While the L2-3 neuron appears to be a part of a network, the L5-6 neuron seems to increase momentarily its firing rate upon reaching a threshold, and then to "saturate", which is suggestive for activation of a neuron not part of its network.

To test whether differential activation takes place in layer II/III vs. layer V/VI, we calculated MUA and SUA before (pre), during (stim) and after (post) ramp stimulation of upper and deeper layers. Similar firing rates were observed (s. table below).

Spiking	E15.5 IUE layer II/III	E12.5 IUE layer V/VI	Significance
MUA pre-stimulus	0.92 ± 0.10	0.97 ± 0.09	0.76
MUA stimulus	3.10 ± 0.31	2.49 ± 0.21	0.14
MUA post-stimulus	2.83 ± 0.53	2.97 ± 0.15	0.72
SUA pre-stimulus	0.42 ± 0.05	0.42 ± 0.04	1.00
SUA stimulus	1.14 ± 0.13	1.08 ± 0.09	0.72
SUA post-stimulus	1.16 ± 0.16	1.21 ± 0.07	0.77

These data indicate that in both experiments a network and not single / few neurons were activated.

In addition, while L2-3 units followed light pulse trains in different frequencies, L5-6 MUAs didn't (lines 219-224). Also, is there a reason why there is no SUA analysis for LV/VI?

The lower ability of layer V/VI neurons to reliably fire when light pulse trains were applied might reflect the higher failure rate observed already at frequencies >16 Hz in vitro (Fig. 2l vs. 2f). We added an explanatory note in the ms (page 8, line 242).

We followed the referee's concern and added SUA analysis for layer V/VI to the manuscript (page 8 and 10, Fig. 3f, Fig. 4h). Moreover, we performed additional patch-clamp experiments from layer V/VI neurons and supplemented the manuscript with novel data and corresponding analyses (page 5-7, Fig. 2, Supplementary Fig. 2).

While the conclusion of the authors that the differences between the different layers reflects a mechanistic difference in involvement in neonatal oscillations may indeed be valid, other interpretations are plausible given the current results. The most severe one is that ChR2 positive

neurons of E15.5 IUE are focally expressed in PL, while those of E12.5 IUE have bigger distances between them and are scattered also to other cortical areas. This may lead to recruitment of a bigger (relative) part of a local network in the 15.5 IUE condition (assuming all experimental conditions, especially fiber diameter and distance from recording site, are the same). In other words- it might be that in the 15.5 IUE condition, a network was activated by light, while in the 12.5 IUE condition just a small fraction of it (or a single neuron in the extreme case) was activated.

To test this possibility, we analyzed the density and number of transfected neurons after 12.5 IUE and 15.5 IUE. While the density was similar (E15.5 Layer II/III: $34.7 \pm 0.8\%$; E12.5 Layer V/VI: $33.1 \pm 1.2\%$ $p=0.288$), the absolute numbers of neurons were indeed different with significantly more transfected neurons in layer II/III (1277.4 ± 36.7 cells/mm²; $p=0.0002$) when compared with layer V/VI E12.5 (1053.3 ± 38.4 cells/mm²). This difference results from the distinct anatomical structure of layers (i.e. more densely packed neurons in upper layers vs. deeper layers (NeuN: LII/III 3698.8 ± 108.7 cells/mm²; LV/VI 3245.2 ± 121.0 cells/mm²; $p=0.007$)). Next, we used the model for propagation of light in brain tissue (Stujenske et al., 2015) to estimate whether the number of activated neurons differs between layers. At a light power sufficient to trigger neuronal firing, 363 neurons in layer II/III and 303 neurons in layer V/VI were activated. These data rule out the possibility that single or few neurons were activated in layer V/VI (page 5 and 8).

Therefore, in order to validate the important message which the authors want to convey, a more comparable expression profile of the two conditions (layer II/III) versus layer V/VI would be more convincing. For this, couldn't one of the transgenic Thy1-ChR2 lines be used in order to achieve a higher specificity for targeting specific layers? From CAMPSALL et al (2002, Developmental Dynamics) 'Characterization of Transgene Expression and Cre Recombinase Activity in a Panel of Thy-1 Promoter-Cre Transgenic Mice' it seems that some lines show already expression of an integrated gene as early as day P14. If this approach is not feasible this alternative should at least be discussed as well as the potential confound mentioned above (network versus small fraction activation).

The additional analyses showed that despite different absolute number of transfected neurons in upper and lower layers, the number of activated neurons is rather similar. The use of suggested transgenic lines would not help to obtain a more comparable expression across layers, since the differences result from anatomical peculiarities of layer II/III vs. layer V/VI. In addition, transgenic lines do not enable area- and layer-specific targeting of neuronal populations, the expression profile in Thy-1 Cre mice described by Campsall et al., 2002 being not restricted to the prefrontal cortex. Consequently, light activation of axonal fibers targeting the PL cannot be excluded in these mice. Moreover, as noted also by the referee, the expression is only detectable at an advanced developmental stage (P14).

Minor remarks:

Line 51 (abstract): "entrainment of neonatal prefrontal networks in fast rhythms critically relies on a local glutamatergic drive...": there is no evidence for that statement in the text. The rhythms might be dependent on di-synaptic GABAergic activation. This claim requires at least pharmacological manipulation of glutamate and/ or GABA receptors.

We rephrased the sentence.

line 107: were other types of adeno viruses used, e.g., would the most commonly used seortypes 2 or 5 work?

According to previous studies (Aschauer et al., 2013; Klein et al., 2006) and preliminary data of co-authors (J.S. Wiegert, C. E. Gee, T. G. Oertner), we tested AAV2/8 because transduction in vitro has been found to lead to efficient transgene expression in neurons already after 1 week. Pseudotypes 2/2 or 2/5 usually show lower expression levels and slower expression onset, which have been confirmed by us in vitro.

line 112: it would be beneficial to include an example figure of the "bad" viral expression, at least in comparison to IUE (especially after devoting a whole paragraph to viruses). Also, a figure about the comparison of CAG and hSyn/Ef1a should be provided.

Viral injection immediately after birth led to no transfection of neurons until P8-10, the age at which the mice were sacrificed. While we visually investigated the brains after using constructs with hSyn and EF1a, we did not image them, but rather sliced the brains with some detectable neurons for patch-clamp recordings. This measure enabled to reduce the number of used animals in line with the EU guidelines (Directive 2010/63/EU). However, these recordings revealed that neurons transfected with ChR2(ET/TC) under the promoter EF1a required much longer light pulses for generating action potentials (see below). Thus, EF1a (and the same was true for hSyn) promotes with weaker efficiency the expression of opsins.

Figure 1. Left, voltage responses of a representative Ef1a-ChR2(ET/TC)-transfected neuron to blue light pulses (473 nm, 5.2 mW/mm²) of 100 and 500 ms duration. Right, bar diagram displaying the mean firing probability of Ef1a-ChR2(ET/TC) transfected neurons in response to blue light pulses of variable duration.

line 137: A positive staining for pyramidal cells, such as CaMKII, should be done.

As suggested, we performed additional staining for CaMKII and showed that all transfected neurons were positive for CaMKII. We added the new data to the manuscript (page 5, Fig. 1d,e,g,h).

Supp fig. 1: not all pups survive the IUE, including the opsin-free treatment, suggesting the treatment is stressful. Therefore, a behavioral comparison to "IUE free" pups would be beneficial.

As suggested, we compared the opsin-transfected pups not only with opsin-free but also to naïve non-electroporated animals. These new experiments revealed similar somatic development of all 3 groups of mice. We supplemented the manuscript with the new data (page 5) and added additional panels to Supplementary Fig. 1g-i.

Line 165: a short explanation for why this measure was taken would be useful (pyramidal cells are non-resonators, so this is another evidence suggesting these are pyramidal cells, or to study if oscillations may be due to intrinsic or network properties)

We added an explanatory note (page 6, line 176).

Line 174: there is a trend towards lower peak amplitude of ChR2 cells. As the number of cells is quite low (5), it will be beneficial to test more cells and exclude the hamper of a different spiking profile.

We extended the in vitro investigation of light-induced spiking by augmenting the number of investigated neurons in layer II/III (n=14) and by adding a similar investigation of neurons in layer V/VI (n=12). The analysis of additional data showed that for layer II/III neurons the peak amplitude is still not significantly different when triggered by light, but the action potential half-width, rise time and decay time are significantly lower when compared with the values from current injection. Of note, layer V/VI pyramidal neurons did not show such decrease (compare Fig. 2e and Fig. 2k). To explain these layer-specific differences between light and current stimulation, we used the Hodgkin-Huxley model. Generally, layer II/III neurons show more immature action potential properties in agreement with the later development of more superficial cortical layers. We modeled the impact of low Na^+ / K^+ conductance (characteristic of immature neurons) and showed that, in line with experimental data (Supplementary Fig. 2e,f), action potentials get faster with increasing current injection only for neurons with low Na^+ / K^+ conductance (Supplementary Fig. 2a-d). Thus, the difference in the timing of action potentials of immature layer II/III neurons in response to current injection and light stimulation most likely results from higher current flowing in response to the opening of the channelrhodopsins.

We added the new data to the manuscript (page 5-7, Fig. 2, Supplementary Fig. 2).

Line 187: for making such a statement testing more frequencies between 16 and 32 Hz would be necessary.

We confirmed the statement by testing additional stimulation frequencies (20, 24, 28 Hz) and we supplemented the ms with the new data (Fig. 2f,l).

Fig. 2: was a similar analysis done for cells in L5-6? (the fig is only about L2-3). If so, are the results comparable?

As previously mentioned, we performed additional experiments and added a similar analysis for layer V/VI transfected neurons after IUE at E12.5 (page 5-7, Fig. 2g-l).

Line 222: AP latency of 6 ms seems quite long for directly activated neurons. From my experience it should be around 3 ms.

The action potential latency for light stimulation mainly depends on the expression strength of the opsin and light intensity used for stimulation. For the light intensity we used in vitro and in vivo (estimated light distribution shown in Suppl Fig 3e) latency of ~6 ms are in agreement with experimental data and computational models for ChR2 (Grossman et al., 2013). With increasing light power the spike delay is reduced (Supplementary Fig. 3d).

Line 243: in order to claim for synchronization in beta, the ISI is not sufficient as the only measurement. It should be accompanied with a phase-lock-analysis. Therefore, this claim should appear later in the manuscript.

We changed the text accordingly (page 9 and 10).

Line 508: it is not clear what "scaled to the immediate downstroke" means

We rephrased this sentence for better understanding (page 16 and 17).

Language and visibility:

I suggest to present the main findings more clearly in one summarizing figure. In the current manuscript version the main information is hidden in subpanels and very hard to extract.

We added a summarizing figure (Fig. 6) to the manuscript.

Supp fig 5 c and f: the arrows are hardly visible (dark contour, for example, would help)

To improve the visibility, we added a dark contour to the arrows in the phase plots (Fig. 4, Supplementary Fig. 6).

Line 316: "led" should be "lead"

Corrected.

References

Aschauer, D. F., Kreuz, S. & Rumpel, S. Analysis of transduction efficiency, tropism and axonal transport of AAV serotypes 1, 2, 5, 6, 8 and 9 in the mouse brain. *PloS one* **8**, e76310, doi:10.1371/journal.pone.0076310 (2013).

Grossman, N. *et al.* The spatial pattern of light determines the kinetics and modulates backpropagation of optogenetic action potentials. *Journal of computational neuroscience* **34**, 477-488, doi:10.1007/s10827-012-0431-7 (2013).

Klein, R. L. *et al.* Efficient neuronal gene transfer with AAV8 leads to neurotoxic levels of tau or green fluorescent proteins. *Molecular therapy : the journal of the American Society of Gene Therapy* **13**, 517-527, doi:10.1016/j.ymthe.2005.10.008 (2006).

Stujenske, J. M., Spellman, T. & Gordon, J. A. Modeling the Spatiotemporal Dynamics of Light and Heat Propagation for In Vivo Optogenetics. *Cell Rep* **12**, 525-534, doi:10.1016/j.celrep.2015.06.036 (2015).

Reviewer #2 (Remarks to the Author):

In this work, Bitzenhofer et al. show that optogenetic activation of layer II/III pyramidal neurons in neonatal mouse prefrontal cortex (PL) can induce network oscillations in beta-gamma frequency range, characteristic patterns of network activity observed in developing cortical circuits. ...

Whether their finding that layer II/III but not layer V/VI pyramidal neurons are the key for generating beta-gamma oscillations during the neonatal period can be generalized to other cortical areas remains an open question.

We supplemented the manuscript with a short discussion of this issue (page 12).

In addition, what is mechanistically different between layer II/III and layer V/VI circuits is not well addressed.

In line with the reviewer's suggestion, we discussed the possible differences between the circuitry of layer II/III and layer V/VI in the light of present and published results (page 12).

However, optogenetic activity manipulation in neonatal mouse brain is technically challenging. This study therefore provides novel and valuable information to many researchers in the field.

We thank the referee for the positive evaluation of the paper as well as for the comments and suggestions that were most helpful.

Specific comments:

1. I have a concern about the area specificity of IUE-mediated gene expression. The authors show only one brain section each for E15.5 IUE and E12.5 IUE (Fig1). In Fig1c (E15.5 IUE), it appears that tDimer2 expression is restricted to the PL. In Fig1f (E12.5 IUE), the expression appears widespread. How about the expression in other areas of other brain sections? If channelrhodopsin is expressed in neurons of other areas and they send axons to the PL, light activation experiments (such as those shown in Fig3 and 4) may be affected because Ch-expressing axons can be easily activated. More specifically, it may be that light activation targeting layer II/III, but not layer V/VI, caused frequency-specific boosting of oscillatory activity because channelrhodopsin expression was restricted to PL in E15.5 IUE but it was widespread in E12.5 IUE. I would like to see the authors provide sufficient information to deny this possibility.

In line with the reviewer's concern, the expression patterns achieved by in utero electroporation unavoidably show a certain amount of variation. In our experiments, 2 out of 9 E15.5 IUE mice and 2 out of 12 E12.5 IUE mice showed a more widespread transfection extending to motor cortex. The remaining pups (7 out of 9 and 10 out of 12, respectively) expressed ChR2(ET/TC) exclusively in the medial PFC. Therefore, it is unlikely that targeting specificity differs for 12.5 and 15.5 IUE. However, we performed additional analyses to determine whether the expression strength after E12.5 and E15.5 IUE correlates with light-induced changes in firing and LFP power. No significant correlations were found for any of the investigated layers or parameters. Moreover, the pups with widespread expression did not show outlying values. We added the new data (page 10) and a new supplementary figure (Supplementary Fig. 8) to the manuscript.

2. In Supplementary Fig. 1d, the rate of successful expression in PL is relatively low. I wonder how the authors chose pups with channelrhodopsin expression in PL for in vivo recording experiments. Did they confirm gene expression in PL by some method before recording, or did they perform recording in all pups then confirm gene expression afterwards? I couldn't find their procedure in the manuscript.

For the first experiments, all animals were recorded and ChR2(ET/TC) expression was confirmed post mortem. For the subsequent experiments, we assessed successful expression in PL using a flashlight to excite fluorescence through the intact skull and skin at P2-3 and we confirmed it in brain slices post mortem. We detailed this issue in Materials and Methods (page 14).

3. In Fig4b legend, does "n=9" mean the number of animals or recordings? If it is the number of animals, could the specific network activity (beta or gamma) be reliably induced in all recording sessions in each animal? If it is the number of recordings, is the data from one animal? This is a question of how reliably could the authors induce beta/gamma band network activity across recording sessions and across animals.

We specified the meaning of n throughout the ms and added details on the animals / cells considered for analysis (legends Fig. 4b,f and Supplementary Fig. 6, 7).

In addition, does the extent of channelrhodopsin expression correlate with the success rate of inducing the specific network activity?

As suggested by the referee, we assessed the relationship between ChR2(ET/TC) expression and success rate of inducing network activity. The results of the new analysis were described in the text (page 10) and Supplementary Fig. 8 (see response to query 1).

4. Page 9, lines 287-291: the authors state that "To examine how light-induced network oscillations spread over PL, we monitored,,,. Augmented beta-band activity was detected,,,". Is the data for this statement presented? I think this is related to the authors' discussion that "During early development, beta oscillations seem to be generated,,. boost their local inhibitory action entraining cortical circuits in gamma frequency band" (page 11, lines 366-369) and important.

We followed the reviewer's suggestion and added a detailed description of activity patterns over upper and deeper layers simultaneously recorded with 4-shank optoelectrodes (page 10-11). The main findings were summarized in an additional main figure and a supplementary figure (Fig. 5 and Supplementary Fig. 9).

5. Page 4, lines 117-118: References are required for "our previous investigations".

We added the corresponding references.

6. Page 5, line 156: does "n=5" mean the number of animals? Slices? Neurons? I thought this was the number of neurons, but found "n=12 neurons" in Fig2b legend and was confused.

We specified the meaning of n throughout the ms.

7. Page 6, line 198: "than" should be "then".

Corrected.

8. Page 9, lines 276-277: the authors state that "did not significantly modify the LFP", but there is no statistical information.

We added the statistical information.

9. In Fig legends of Fig3, 4, SuppleFig4, SuppleFig5, SuppleFig6, the information about which test was used is missing.

For all figures we specified statistical tests used in the legends.

10. In Fig4b and Fig4f, boxes with dotted lines show the period before light stimulation. Is this 1.5s or 3s? In Fig4cd and Fig4g, do boxes with solid lines show 3s? How about the boxes in SuppleFig6b? In SuppleFig6b legend, does "n=10" mean the number of recordings or animals?

We specified the duration of time window in the figure legend. Generally, the boxes correspond to 1.5 s. Phase locking analyses illustrated in Fig. 4C, d, g, h required high number of spikes and consequently, the size of time window was augmented to 3 s, taking into account that at neonatal age the firing rates are low.

As previously mentioned, we specified the meaning of n throughout the ms.

Reviewer #3 (Remarks to the Author):

This study uses a combination of different methods to elucidate the cellular and network mechanisms underlying beta-gamma activity in the prefrontal prelimbic (PL) cortex of P8-P10 mice. The main result is shown in figure 4: layer 2/3 (Fig 4b), to a minor extent layer 5/6 (Fig 4f) pyramidal neurons boost beta-gamma, to some extent also theta network activity.

We thank the referee for the comments and suggestions. We hope that the large amount of data and analyses as well as the extensive discussion in the revised manuscript strengthen the novelty and impact of our study.

The authors already previously uncovered the cellular interactions contributing to theta and beta-gamma oscillations in P6-P8 mice using similar recording techniques and identified L5 pyramidal cells and GABAergic interneurons as critical components to generate this network activity (Bitzenhofer et al., 2015).

The important role of L2/3 in generating spontaneous theta- and beta/gamma oscillations in developing cerebral cortex (>P8) has been already documented previously (Gireesh and Plenz, 2008). This paper is not even cited. Therefore a statement as "we provide the first causal demonstration that cortical oscillations during early development can emerge as result of cell type-specific activation within local intracortical circuits" (line 308) is not true.

*We agree with the reviewer that previous data from our and other labs aimed at elucidating the neonatal circuitry contributing to early oscillatory rhythms. Using extra- and intracellular recordings, these studies gave first insights into the wiring scheme of developing cortex. However, none of them provided **causal** evidence how a specific neuronal population contributes to the generation of neonatal rhythms; the conclusions of the previous studies rather rely on temporal correlations between population and individual neuronal activity. It was necessary to use optophysiology in the neonatal brain to establish causal links between neuronal populations and activity patterns. As also highlighted by Reviewer #1 and #2, the present study is the first that achieved this goal. Similarly, in the adult brain the role of interneurons in the generation of gamma activity was only unequivocally demonstrated by optogenetic manipulation (Cardin et al., 2009), even though this role was proposed by many earlier studies.*

We apologize for omitting the study by Gireesh and Plenz, it is now cited in the ms (page 3).

The ms contains many misleading or even false statements. I just give a few examples from the first pages.

Since most of the concerns do not reflect conceptual mistakes, we substantially rephrased the text to avoid misleading statements.

- In title and text authors use the term "neonatal", but mice older than P8 were studied. Considering the rapid development and the general terminology in rodent development, the term neonatal should not be used.

To our knowledge (and regret!) no generally accepted terminology for different developmental stages is available. The stage of maturity varies over brain areas, the PFC being one of the regions with delayed development. In previous studies we defined the "neonatal period" in limbic circuits as time-window during which animals lack sensory activation (retina is light insensitive, ears are closed, low motor activity) and the activity patterns in the investigated area are discontinuous (i.e. network oscillations alternate with periods of "silence"). To facilitate the comparison of present and previous data we kept the terminology "neonatal" in the ms. To avoid

misunderstanding, we clearly specified, which age has been classified as neonatal (Introduction, page 3).

- First lines of Introduction: "For long time the developing brain has been considered as the premature form of the adult one. .. This dogma has been profoundly challenged". As we can read in every text book on developmental neurobiology from the last 40-50 years, the developing brain is of course not a premature form of the adult brain. Many processes are unique for the early development. This "dogma" simply does not exist.

We rephrased the text (page 3).

- Line 68: "the depolarizing but inhibitory action of GABA-A": This is a very selective and incorrect form of citing the literature of the last 20 years from many labs.

While GABA action in the developing brain exceeds the focus of the present ms, we added as reference a recent review (Ben-Ari, 2014) that summarizes the abundant literature on this topic of the last decades. The pioneering study of Kirmse and colleagues published last year in Nature Comm. is an important piece of experimental evidence for the depolarizing but shunting inhibitory action of GABA. In line with the referee's concern, we rephrased the sentence.

- Line 68: "the directed interactions ensuring both local and long range network coupling are exclusively present during early postnatal development." Of course local and long-range network coupling is strongly expressed in the adult brain.

We rephrased the text for better understanding (page 3, line 69).

- Line 71: "Recently, specific neuronal sub-populations and their transient connectivity have been identified as being critical for proper maturation of neuronal circuits". Subplate cells, as one or maybe the most important population of transient cells in the cerebral cortex and their connectivity have been described by Shatz and others not so "recently" (e.g. McConnell et al., 1989; Luskin and Shatz, 1985).

We rephrased the sentence for better understanding (page 3, line 70).

The ms also contains meaningless statements, such as "neuronal circuits as pre-adapted template of future behavioral requirements" (line 40).

We rephrased this statement (page 3, line 46).

About 70-80% of this study describes the methods (Results page 4-8, Figs. 1-3). Only the last chapter in Results on page 8f and Fig. 4 is related to the main question of this study. Looking at Fig 4 as the main result of this study I find the conclusion of this paper not convincing.

We agree with the reviewer that one of the main strength of the study is the methodological development (i.e. establishment of protocols for optogenetic manipulation of neonatal circuits) that represents the prerequisite for causal interrogation of developing networks. However, the manuscript is NOT a mere description of methods but equally provides knowledge gain (see 3 out of 4 sub-sections of the study and the corresponding figures) that has been considered by Reviewer #1 and #2 as "valuable information to many researchers in the field". Moreover, we supplemented the revised ms with a large amount of novel data (e.g. properties of light-induced action potentials in layer V/VI pyramidal neurons, oscillatory coupling across cortical depth revealed by 4-shank optoelectrodes) that reinforce the conclusions of the paper.

In their previous paper (Bitzenhofer et al., 2015) the authors also studied awake animals. This study would be more valuable if recordings also would have been done in non-anesthetized animals.

The study cited by the reviewer showed that the activity patterns recorded in asleep non-anesthetized and urethane-anesthetized animals are indistinguishable. Therefore, the benefit of using non-anesthetized mice is not obvious when investigating the resting-state activity in developing brain. Moreover, we observed that the recording procedure in head-fixed mice in the absence of anesthesia is more stressful at this young age, since they cannot be trained for the procedure. For these reasons and in line with the EU guidelines for minimizing discomfort in animals (Directive 2010/63/EU), we decided to achieve the study under urethane anesthesia.

The behavioral tests used in this study are certainly not related to the transfected brain region. Reflexes are driven by spinal cord and not prefrontal cortex. Therefore a statement as "To exclude non-specific effects of the transfection with ChR2" is misleading (line 146).

We rephrased the text for avoiding this misunderstanding. Testing somatic development and reflexes were not meant to prove the intact prefrontal function but to assess whether the manipulation (IUE, transfection with light-sensitive proteins, etc.) impairs the overall development of pups and inevitably, the brain function.

line 328: "The preset study ... identifies one generator of neonatal network activity" is misleading. The "generator" maybe the presynaptic network activating L2/3 neurons in PL, but this network has not been identified.

Following the reviewer's concern, we rephrased the sentence.

Last line abstract: "This approach enables the interrogation of developing circuits and their later behavioral readout." But behavioral readout has been studied in P2-P8 animals only (line 411).

We rephrased for avoiding misunderstanding. This sentence is meant to give an outlook of the benefits of the established method. It does not refer to the testing of somatic development and reflexes, but to later juvenile-adult cognitive behavior.

The main result of this study maybe trivial, namely that L2/3 pyramidal cells tend to fire in beta/low gamma frequency more strongly than L5/6 pyramidal cells. Unfortunately this question is not clearly addressed (e.g. Fig. 2) despite the fact that previous studies support this hypothesis (e.g. Wespatat et al., 2004;Gireesh and Plenz, 2008).

As mentioned by the reviewer, previous studies set the hypothesis that layer II/III neurons contribute to early oscillatory activity in beta frequency band, yet the direct experimental proof has been obtained in the present study. Not only layer II/III neurons preferentially fire in this frequency band, but their stimulation augmented the power, phase-locking and coherence of these oscillations. This was not the case for layer V/VI pyramidal neurons. The new experiments using 4-shank optoelectrodes spanning both upper and lower layers of PL confirm these data (Fig. 5, Supplementary Fig. 9).

To get a better understanding of the network activity, analyses of simultaneous local field potential recordings in upper and lower layers in addition to SUA and MUA data would be most valuable. This is very easy to do and gives more information on the local network activity than some data shown in the paper.

We agree that simultaneous recordings from upper and lower cortical layers would strengthen the main conclusions of the study. As mentioned above, we complemented the already available LFP recordings with additional 4-shank recordings and analyzed in more detail the organization

and synchrony of network oscillations over all prelimbic layers. We added the new findings to the text (page 10-11) and two new figures to the ms (Fig. 5 and Supplementary Fig. 9).

I doubt that the increasing ramp stimulation used in Fig. 3 and 4 over several seconds is very physiological. Which physiological synaptic input would provide such an activation of pyramidal cells in PL?

The use of ramp stimulation, which has been previously used in numerous studies (Akam et al., 2011; Adesnik and Scanziani, 2010; El Hady et al., 2013), has been preferred because it leads to an increased firing of pyramidal neurons for the duration of naturally occurring spindle bursts and nested gamma-spindle bursts (i.e. 2-3 s), the major network oscillations identified in the neonatal PFC. Moreover, it does not produce artificially synchronous firing of pyramidal neurons as resulting from stimulation with trains of short light pulses.

References

Adesnik, H. & Scanziani, M. Lateral competition for cortical space by layer-specific horizontal circuits. *Nature* **464**, 1155-1160, doi:10.1038/nature08935 (2010).

Akam, T., Oren, I., Mantoan, L., Ferenczi, E. & Kullmann, D. M. Oscillatory dynamics in the hippocampus support dentate gyrus-CA3 coupling. *Nat Neurosci* **15**, 763-768, doi:10.1038/nn.3081 (2012).

Ben-Ari, Y. The GABA excitatory/inhibitory developmental sequence: a personal journey. *Neuroscience* **279**, 187-219, doi:10.1016/j.neuroscience.2014.08.001 (2014).

Cardin, J. A. *et al.* Driving fast-spiking cells induces gamma rhythm and controls sensory responses. *Nature* **459**, 663-667, doi:10.1038/nature08002 (2009).

El Hady, A. *et al.* Optogenetic stimulation effectively enhances intrinsically generated network synchrony. *Front Neural Circuits* **7**, 167, doi:10.3389/fncir.2013.00167 (2013).

Kirmse, K. *et al.* GABA depolarizes immature neurons and inhibits network activity in the neonatal neocortex in vivo. *Nat Commun* **6**, 7750, doi:10.1038/ncomms8750 (2015).

Reviewers' comments:

Reviewer #1 (Remarks to the Author):

The authors provided satisfactory answers to my previous concerns, added data and analysis and improved the manuscript significantly.

However, there is one major point I'm still concerned about:

Fig. 2, f compared to I: adding the responses of layer V/VI cells to light pulses in different frequencies really helps. However, although it clarifies the difference in response to 16 Hz, it seems like layer V/VI neurons are actually more capable to follow higher frequencies (especially true for 64 Hz). At least this is implied by the exemplary neuron in this figure. Therefore, it is still not clear why the authors excluded the deeper layers from analysis of pulse trains, as they state in line 242 (for example, in supp. figures 4 and 7 only layer II/III are presented, and also in the main text, line 312-315).

Minor comment:

Typo on Line 336: change "To pieces..." into "Two pieces..."

Reviewer #2 (Remarks to the Author):

The authors have gone to great lengths to address all the comments and concerns. I am happy with their revisions. Just a typo: "Two" instead of "To" in page 10 line 336.

Reviewer #3 (Remarks to the Author):

no comment

Reviewer #1 (Remarks to the Author):

The authors provided satisfactory answers to my previous concerns, added data and analysis and improved the manuscript significantly.

We thank the referee for the comments and suggestions that substantially improved the manuscript.

However, there is one major point I'm still concerned about:

Fig. 2, f compared to l: adding the responses of layer V/VI cells to light pulses in different frequencies really helps. However, although it clarifies the difference in response to 16 Hz, it seems like layer V/VI neurons are actually more capable to follow higher frequencies (especially true for 64 Hz). At least this is implied by the exemplary neuron in this figure. Therefore, it is still not clear why the authors excluded the deeper layers from analysis of pulse trains, as they state in line 242 (for example, in supp. figures 4 and 7 only layer II/III are presented, and also in the main text, line 312-315).

As suggested by the referee, we tested whether neurons in deeper layers are more capable to follow high frequencies of pulse stimulation when compared with neurons in the upper layers. Comparison of firing probability for all frequencies of stimulation revealed that layer II/III neurons follow significantly ($p=0.004$) better the 16 Hz light pulses than layer V/VI neurons, whereas no significant differences in the firing probability were detected for light pulses at 20, 24, 28, 32 and 64 Hz. To mirror these results, we replace the misleading example shown in Fig. 2l. We added the new analysis to the manuscript (page 7, lines 208-216).

Minor comment:

Typo on Line 336: change "To pieces..." into "Two pieces..."

We corrected the typo in the text.

Reviewer #2 (Remarks to the Author):

The authors have gone to great lengths to address all the comments and concerns. I am happy with their revisions. Just a typo: "Two" instead of "To" in page 10 line 336.

We thank the referee for the helpful comments.

We corrected the typo in the text.

Reviewer #3 (Remarks to the Author):

REVIEWERS' COMMENTS:

Reviewer #1 (Remarks to the Author):

The authors addressed my remaining concern sufficiently.